# Structural basis of rapid actin dynamics in the evolutionarily divergent *Leishmania* parasite

Tommi Kotila [1], Hugo Wioland [2], Muniyandi Selvaraj [1], Konstantin Kogan [1], Lina Antenucci[1], Antoine Jégou [2], Juha T. Huiskonen [1], Guillaume Romet-Lemonne[2] & Pekka Lappalainen [1✉]

Actin polymerization generates forces for cellular processes throughout the eukaryotic kingdom, but our understanding of the 'ancient' actin turnover machineries is limited. We show that, despite >1 billion years of evolution, pathogenic *Leishmania major* parasite and mammalian actins share the same overall fold and co-polymerize with each other. Interestingly, *Leishmania* harbors a simple actin-regulatory machinery that lacks cofilin 'cofactors', which accelerate filament disassembly in higher eukaryotes. By applying single-filament biochemistry we discovered that, compared to mammalian proteins, *Leishmania* actin filaments depolymerize more rapidly from both ends, and are severed > 100-fold more efficiently by cofilin. Our high-resolution cryo-EM structures of *Leishmania* ADP-, ADP-Pi- and cofilin-actin filaments identify specific features at actin subunit interfaces and cofilin-actin interactions that explain the unusually rapid dynamics of parasite actin filaments. Our findings reveal how divergent parasites achieve rapid actin dynamics using a remarkably simple set of actin-binding proteins, and elucidate evolution of the actin cytoskeleton.

[1] Institute of Biotechnology and Helsinki Institute of Life Science, University of Helsinki, P.O. Box 56, 00014 Helsinki, Finland. [2] Université Paris Cité, CNRS, Institut Jacques Monod, F-75013 Paris, France. ✉email: pekka.lappalainen@helsinki.fi

Flagellated eukaryotic parasites of the *Leishmania* genus infect various vertebrate and invertebrate hosts, and a subset of *Leishmania* species cause severe diseases in humans[1]. Moreover, related *Trypanosoma* parasites are a major health burden by causing various trypanosomiasis, including "African sleeping sickness". Pathogenic *Leishmania* species need both sandfly and mammalian hosts for their life cycle, and exist in two forms: non-motile oval amastigotes that reside and multiply in mammalian macrophages, and flagellated, motile promastigotes that migrate to the salivary glands of the sandfly, and can be transmitted to the mammalian host during a blood meal. Due to their peculiar life cycle, and >1 billion years of distance in evolution, the cell biology of *Leishmania* parasites exhibits notable differences compared to animal cells[2,3].

Actin is among the most highly conserved proteins in eukaryotes, and its central role in cells puts it under tremendous evolutionary pressure. Comparing actins and actin-regulatory machineries of very distant organisms is particularly insightful, as it allows addressing questions about the evolution of the cytoskeleton. In animal cells, the actin cytoskeleton is important for motility, morphogenesis, endocytosis, and organelle dynamics. In endocytosis and cell migration, coordinated polymerization of actin filaments at their rapidly growing barbed ends against the plasma membrane provides force for membrane deformation. The rapid actin filament assembly must be balanced by filament disassembly to maintain the supply of polymerization competent actin monomers. The dynamics of isolated animal actin filaments are slow, and thus an array of proteins evolved to accelerate actin dynamics and control the architecture of the actin cytoskeleton[4,5]. The actin cytoskeleton is also conserved in trypanosomatids (*Leishmania* and *Trypanosoma* parasites), but it exhibits major differences compared to the extensively-studied metazoan and yeast actin cytoskeletons. While yeast and animal actin orthologs display ~90% identity to each other at the amino acid level, the *Leishmania* and *Trypanosoma* actins are more divergent and display only ~70% identity to animal actins[6]. Genetic and microscopy studies suggest that in trypanosomatids actin associates with endosomal structures at the flagellar pocket, and contributes to endocytosis[7–9]. However, there is no structural information from *Leishmania* or *Trypanosoma* actins, and thus their possible differences to animal actins are not known. Interestingly, a biochemical study carried out on purified His-tagged *Leishmania donovani* actin suggested that individual filaments rapidly bundle[10].

Metazoan and yeast cells harbor a large array of actin-regulating proteins, which control actin filament nucleation, polymerization, and disassembly, as well as regulate the cytoplasmic actin monomer pool. On the other hand, *Leishmania* species display a much simpler actin-regulatory system consisting of <10 canonical proteins[11]. These include a single isoform of actin-depolymerizing factor (ADF)/cofilin, which in "higher" eukaryotes promotes actin turnover by severing filaments and affecting filament dynamics at both ends[12–14], the Arp2/3 complex and formins, which nucleate actin filaments[15], twinfilin, which regulates filament barbed end dynamics[16], actin filament binding protein coronin[17], as well as profilin and cyclase-associated protein (CAP), which regulate the actin monomer pool[18]. Based on amino acid sequences, and a small number of genetic and biochemical studies, these proteins display notable differences to their animal orthologs[19–23]. Moreover, certain key regulators of actin dynamics are absent in *Leishmania*. These include Capping protein, which is a central regulator of filament barbed end assembly in metazoan organisms and yeasts, and Actin-interacting protein 1 (Aip1), which is an important cofactor of ADF/cofilin-promoted filament severing[24–27]. Moreover, the *Leishmania* parasites express a short version of CAP, which is composed of C-terminal domains that 'recharge' actin monomers with ATP[28,29], but lacks the N-terminal domain that promotes pointed end depolymerization of ADF/cofilin-decorated actin filaments[28,30,31]. Because the acceleration of ADF/cofilin-mediated actin filament severing by Aip1 and promotion of filament pointed end depolymerization by CAP are critical for rapid turnover of animal and yeast actin filaments[32–34], it is unclear how *Leishmania* can maintain rapid actin dynamics in the absence of these key factors. Moreover, due to the lack of structural information on *Leishmania* or *Trypanosoma* actins, and their complexes with actin-binding proteins, our understanding of the similarities and differences of actin-regulatory mechanisms between metazoan organisms and evolutionarily distant flagellated parasites is scarce.

In this work, we focused on the *Leishmania major* parasite to elucidate the evolution of actin and actin-regulatory machinery, as well as to uncover how rapid actin dynamics can be achieved in divergent flagellated parasites in the absence of key actin-binding proteins. We show that *L. major* actin filaments undergo much more rapid depolymerization at both barbed and pointed ends compared to animal actins, and reveal that *Leishmania* ADF/cofilin globally fragments actin filaments >100-fold more efficiently than mammalian ADF/cofilin. By determining the cryo-EM structures of bare and ADF/cofilin-decorated *Leishmania* actin filaments, we uncover the underlying molecular principles. Collectively, this study reveals how *Leishmania* parasites maintain actin filament turnover in the absence of central actin filament disassembly proteins.

## Results

**Evolutionary conservation and polymerization L. major actin.**
Actin has maintained remarkably high sequence conservation across the eukaryotic kingdom. For example, budding yeast and human actin orthologs display ~90% sequence identity despite their large evolutionary distance (Fig. 1a, b). The most divergent eukaryotic actin is from *Giardia*, which appears to lack all canonical actin-binding proteins[35]. The most distant eukaryotes expressing actin and canonical actin regulators are the flagellated trypanosomatids parasites (*Leishmania* and *Trypanosoma* species). *L. major* actin (LmActin) exhibits only ~70% amino acid sequence identity to biochemically well-characterized actins from budding yeast, rabbit, or human. It is also worth noting that LmActin is highly diverged from the malaria actin[36–40]. The ancestral actin ortholog from *Lokiarchaeum* is equally distant from all aforementioned actins (Fig. 1a, b).

To study the characteristics of divergent trypanosomatid parasite actin, we synthesized a codon-optimized LmActin cDNA for baculovirus insect cell expression and purified it as a C-terminally His-tagged β-thymosin fusion protein[41]. The His-tag-β-thymosin fusion was cleaved with chymotrypsin[42] (Supplementary Fig. 1a), and this resulted in actin without additional residues in the C-terminus (see Methods). We first studied the polymerization kinetics of LmActin by microfluidics-coupled TIRF microscopy[43]. In this setup, actin filaments are polymerized from surface-anchored spectrin-actin seeds, exposed sequentially to different solutions of actin and regulatory proteins controlled by microfluidics. Filaments do not interact with the surface except at the anchored end, and they thus align parallel to the flow direction (Fig. 1c and see Methods). To overcome potential artefacts that could be caused by direct labeling, we applied three different strategies (Fig. 1c). In the first strategy, we elongated filaments from spectrin-actin seeds exposed sequentially to labeled rabbit skeletal muscle α-actin (RbActin) and unlabeled LmActin, yielding segmented, micrometers long single actin filaments. Despite the large evolutionary distance between the two

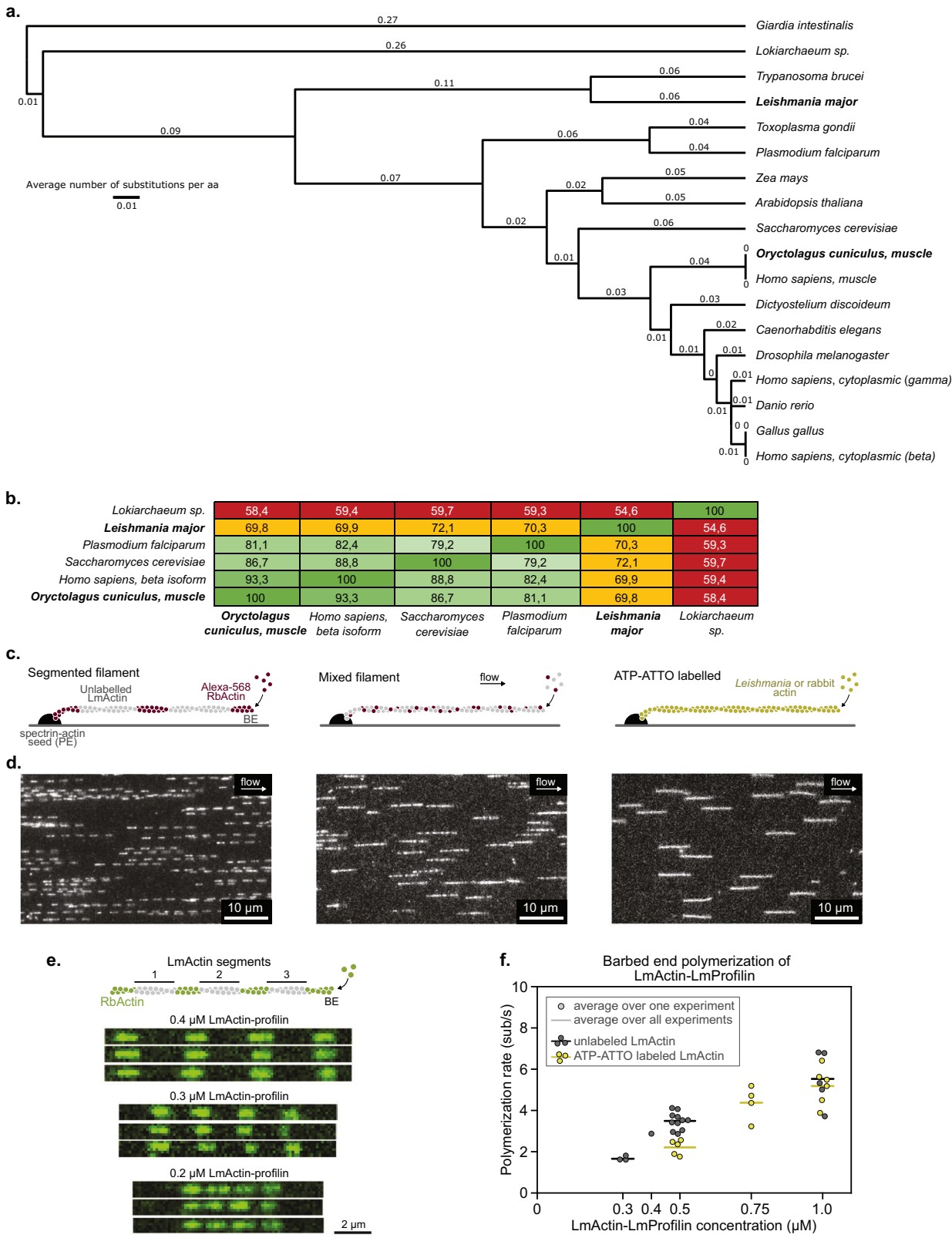

actins, mammalian and *Leishmania* actins could co-polymerize with each other. As a second strategy, we mixed a small amount of labeled RbActin with LmActin. Similarly, they co-polymerized into single homogeneous filaments. As the third strategy, we polymerized LmActin loaded with fluorescent ATP-ATTO-488 prior to polymerization, following the method by ref. [44]. Also, this approach resulted in long LmActin filaments (Fig. 1d).

In the "segmented" polymerization studies, we noted that the unlabeled LmActin segments varied in lengths. More careful analysis revealed that the polymerization rate of LmActin decreases over time, especially at higher LmActin concentrations, indicating possible spontaneous nucleation in our actin stock that would decrease the G-actin pool. To maintain a pool of polymerizable LmActin monomers for longer periods of time,

**Fig. 1 Polymerization kinetics of the divergent *Leishmania major* actin. a** Phylogenetic tree of selected evolutionarily divergent actins based on amino acid sequences. *Leishmania major* parasite actin (LmActin) and α-skeletal muscle rabbit actin (*Oryctolagus cuniculus*, RbActin) used in this study are highlighted in bold. See the Source Data file for the sequence accession codes from Uniprot. **b** Pairwise comparison of the sequence identity between selected actins. **c** Methods applied to assemble and visualize LmActin filaments in microfluidics-coupled TIRF microscopy experiments. BE barbed end, PE pointed end. **d** Examples of observed actin filaments assembled using the methods described in panel **c**. **e** Typical "segmented" filaments assembled by alternating between a solution of Alexa-labeled RbActin (green) and a solution containing equimolar concentrations of unlabeled Lm-G-actin and LmProfilin. For each filament, the three LmActin segments were polymerized for 5 min with the same solution containing either 0.2, 0.3, or 0.4 μM LmActin-LmProfilin. With 0.2 μM LmActin-LmProfilin, the resulting segments were not long enough to measure the polymerization rate accurately. **f** Quantification of the barbed end polymerization rates with equimolar LmActin and LmProfilin, either unlabeled (in segmented filaments, gray) or labeled with 1–4 μM ATP-ATTO-488 (yellow). The polymerization rate increases with LmActin:LmProfilin concentration. Please note that especially at higher profilin:actin concentrations there is some divergence in the polymerization rates between individual experiments, and this is most likely due to spontaneous nucleation of LmActin in solution. Each data point represents an average over all measured filaments (*n*) from independent experiments (*N*), and lines correspond to the average over all experiments. For unlabeled LmActin: 0.3 μM $N = 3$, $n = 60, 60, 60$; 0.4 μM $N = 1$, $n = 60$; 0.5 μM $N = 11$, $n = 50, 40, 40, 50, 60, 20, 20, 20, 20, 20, 10$; 1 μM $N = 5$, $n = 20, 20, 30, 40, 40$. For ATP-ATTO labeled LmActin: 0.5 μM $N = 5$, $n = 20, 20, 20, 20, 30$; 0.75 μM $N = 4$, $n = 20, 20, 20, 20$; 1 μM $N = 6$, $n = 30, 30, 20, 20, 16, 20$.

we mixed LmActin with equal concentrations of *Leishmania* profilin (LmProfilin) and learned that profilin indeed prevented spontaneous filament nucleation (Fig. 1e and Supplementary Fig. 1b–d). Based on experiments with different concentrations of LmActin and equimolar concentrations of *Leishmania* profilin, we estimated a barbed end polymerization rate constant of ~6 subunits/s/μM (Fig. 1f and Supplementary Fig. 1e). These results suggest that LmActin barbed ends polymerize more slowly than what has been reported for RbActin[45], yielding a critical concentration of ~0.1–0.2 μM. We also noted that ATP-ATTO-loaded LmActin monomers polymerized roughly 25% more slowly compared to native LmActin monomers, highlighting the importance of testing the effects of fluorescent probes on actin dynamics (Fig. 1f and Supplementary Fig. 1e). Moreover, solutions containing a mixture of *Leishmania* and mammalian actin exhibited a slower polymerization rate than solutions with either *Leishmania* or mammalian proteins, suggesting weaker interaction between LmActin and RbActin subunits (Supplementary Fig. 1f). Please note that the pointed end polymerization rate of LmActin cannot be determined with these approaches, due to spontaneous nucleation of LmActin in the absence of profilin and because profilin inhibits filament pointed end elongation. Together, these experiments demonstrate that *L. major* actin is capable of polymerizing into long filaments. Moreover, despite relatively low sequence identity, parasite and animal actins are able to co-polymerize.

***Leishmania* actin filaments depolymerize rapidly from both ends**. Next, we examined the depolymerization dynamics of LmActin filaments at their barbed ends. Strategies utilizing both segmented and ATP-ATTO-labeled filaments revealed that "aged" LmActin filaments underwent rapid depolymerization in the absence of free monomers. This corresponds to an average rate of ~25–40 subunits/s for ADP-actin filaments from experiments performed on different days, with different actin preparations and experimental setups (Fig. 2a, b and Supplementary Fig. 2a). This is ~5-fold more rapid than barbed end depolymerization of ADP-RbActin filaments under identical experimental conditions (Fig. 2a, b). Interestingly, experiments carried out with 15% Alexa488-labeled RbActin monomers and 85% unlabeled LmActin resulted in uniform incorporation of the fluorescently-labeled RbActin subunits in the filaments, which depolymerized on average even more rapidly, up to ~40 subunits/s (Supplementary Fig. 2a). This suggests that LmActin is either in a different conformational state compared to RbActin in the filament, or that nonoptimal intra-strand contacts promote more rapid monomer dissociation from the barbed ends of the "mixed" actin filaments.

We next examined the effects of nucleotide state on the barbed end depolymerization of LmActin filaments. By "forcing" actin filaments into the ADP-Pi state with a high phosphate concentration in the buffer, we measured barbed end depolymerization rates of ~6 subunits/s and ~1 subunits/s for LmActin and RbActin, respectively (Fig. 2c). Moreover, by switching the "high phosphate" buffer to a buffer without phosphate allowed us to monitor the slow acceleration of filament depolymerization indicative of phosphate release (Supplementary Fig. 2b, c). From these experiments, we estimated that the rate of Pi-release from actin filaments is ~5–10-fold more rapid in LmActin compared to RbActin under the same experimental conditions (Fig. 2d). Thus, similar to RbActin, the release of phosphate triggers the faster departure of actin monomers from the filament barbed ends, but the lifetime of the ADP-Pi state in LmActin filaments is greatly reduced compared to mammalian actin.

Finally, we measured the pointed end depolymerization dynamics of LmActin filaments. We loaded LmActin filaments with ATP-ATTO and immobilized the filaments on the surface of the microfluidic chamber with biotin-anchored human gelsolin. These experiments revealed a pointed end depolymerization rate of ~3.5 subunits/s for LmActin. This is ~20-fold more rapid compared with the pointed end depolymerization rate of RbActin (Fig. 2e, f). Interestingly, experiments performed by mixing 20% of Alexa488-RbActin and 80% LmActin reduced the pointed end depolymerization of LmActin by threefold, whereas 9% Alexa488-RbActin fraction showed an almost identical rate to experiments with ATP-ATTO loaded LmActin (Supplementary Fig. 2d). This suggests that a higher fraction of RbActin can transiently "cap" the pointed ends of filaments, and thus reduce the total depolymerization rate of LmActin filaments.

Collectively, these experiments demonstrate that LmActin filaments, regardless of the nucleotide state, exhibit more rapid depolymerization dynamics at both ends compared to mammalian actin. Moreover, the faster phosphate release kinetics further enhances the overall rapid disassembly of LmActin filaments.

**Cryo-EM structures of *Leishmania* ADP- and ADP-Pi-actin filaments**. To uncover the molecular basis of rapid LmActin filament depolymerization, we determined the structures of ADP- and ADP-Pi-states of LmActin by cryogenic electron microscopy (cryo-EM) in the absence of any actin-stabilizing agents at average resolutions of 2.7 and 3.3 Å, respectively (Fig. 3a, b, Supplementary Fig. 3a, and Supplementary Table 1). Helical parameters of LmActin were very similar to the ones reported for F-actin structures from other species (Supplementary Table 1)[46–48]. Overall, these structures of *Leishmania* actin demonstrate that the helical nature of actin is remarkably

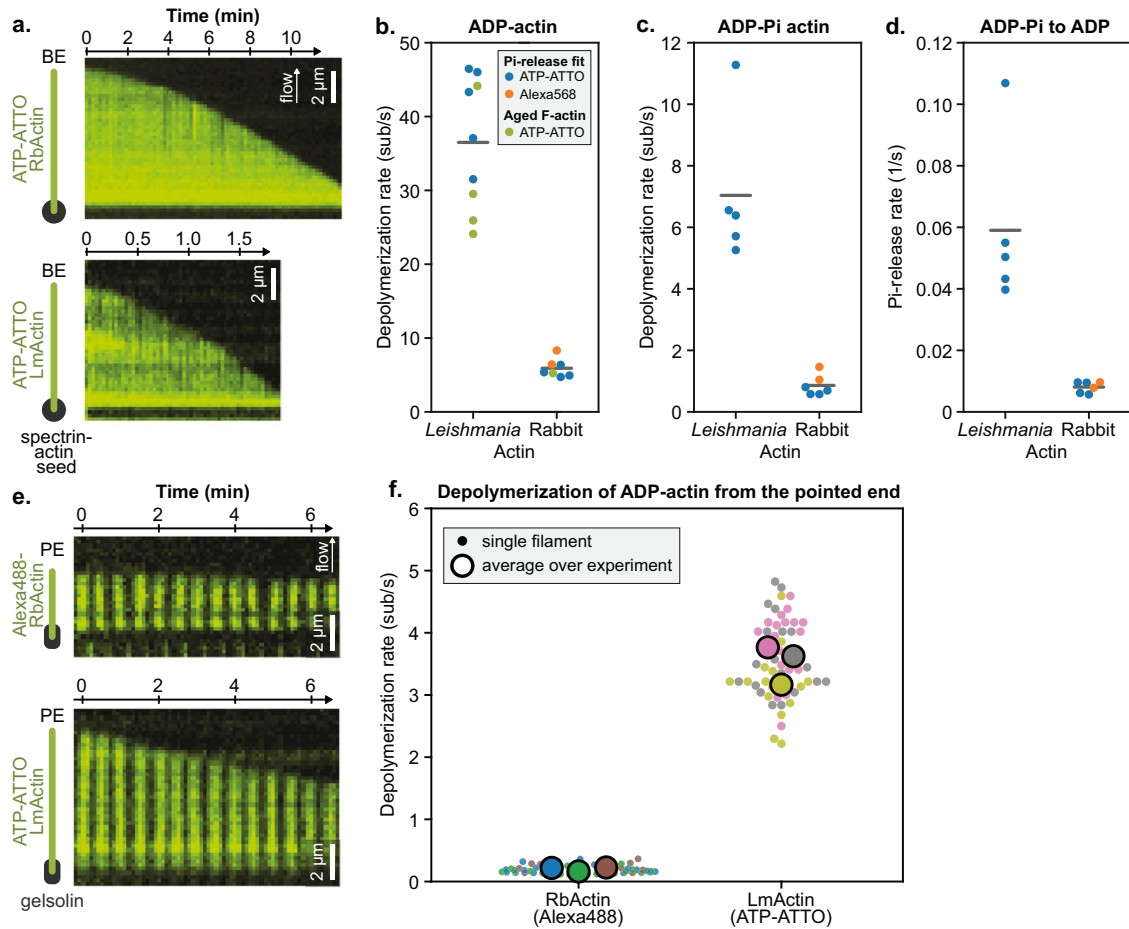

**Fig. 2** ***Leishmania* actin filaments undergo rapid depolymerization from both ends. a** Typical kymographs of RbActin and LmActin filaments (visualized using ATP-ATTO-488) depolymerizing from their barbed ends. Actin subunits bear an ADP-Pi nucleotide at time $t = 0$, which is released over time to result in ADP-actin filament, as observed by the acceleration of the depolymerization rate. Note the difference in time scales between the RbActin and LmActin kymographs. **b–d** Comparison of the barbed end depolymerization rates of *Leishmania* and rabbit muscle ADP-actins (panel **b**) and ADP-Pi-actins (panel **c**), and the Pi-release rates of LmActin and RbActin filaments (panel **d**). Green symbols represent an average depolymerization rate of "aged" actin filaments. Orange and blue symbols represent values obtained by fitting the depolymerization rate of Pi-loaded actin filament vs. time over different data points of one movie (see Methods "Pi-release fit", see Supplementary Fig. 2b, c). Note that the RbActin labeling strategy, with either ATP-ATTO or Alexa568, does not affect the Pi-release rate. Each data point represents an average over all measured filaments ($n$) from independent experiments ($N$), and lines correspond to the average over all experiments. $N = 5$ and $n = 71, 25, 73, 58, 85$ for *Leishmania* actin Pi-release; $N = 4$ and $n = 81, 87, 60, 60$ for rabbit actin Pi-release fit with ATP-ATTO; $N = 2$ and $n = 60, 60$ for rabbit actin Pi-release fit with Alexa568; $N = 4$ and $n = 20, 20, 40, 40$ for aged *Leishmania* F-actin; $N = 1$, $n = 20$ for aged rabbit F-actin. **e** Typical kymographs of RbActin (labeled with Alexa488) and LmActin (ATP-ATTO) filaments depolymerizing from their pointed ends. Filaments were pre-polymerized in solution for >30 min and were thus composed of ADP-actin (see Methods). **f** Pointed end depolymerization rates of LmActin and RbActin actin. Each color depicts an independent experiment. Small data points describe the depolymerization of all measured filaments ($n$), and large ones represent averages over independent experiments ($N$). RbActin: $N = 3$ and $n = 20, 20, 10$. LmActin: $N = 3$ and $n = 20, 20, 16$.

conserved from divergent unicellular parasites to mammals. The conformational state of *Leishmania* ADP-actin subunits within filaments is also very similar (RMSD 0.785 Å) to that of muscle actin, but with small variations in the D-loop position and in the position of subdomain 4 at the pointed end (Fig. 3c). Moreover, the subunit conformation in the ADP-Pi state is nearly identical to the reported ADP-Pi state of muscle actin (RMSD 0.454 Å) (Fig. 3d). Thus, the overall actin fold and conformations in different nucleotide states across evolution appear well-conserved.

A comparison of sequence conservation between *Leishmania* actin and rabbit skeletal muscle actin (used in the biochemical studies) shows differences, especially in the regions that are critical for both inter- and intra-strand filament contacts. These regions include the D-loop, the H-plug, amino acids 307–327 in subdomain 3, as well as residues 193–201 and 228–252 at the

pointed end face of subdomain 4 (Fig. 3e and Supplementary Fig. 3b). Examining the subunit interfaces of actin filament in more detail reveals an interesting set of amino acid substitutions in the H-plug region, which is buried in the center of filament, and stabilizes the filament core together with the D-loop and C-terminus from the two neighboring subunits (Fig. 3f, g). In LmActin, the subdomain 3 border of the H-plug has two substitutions (His275Pro and Ala272Pro), which most likely limit the flexibility of this region and its movement towards the center of the filament. The connecting loop at the subdomain 4 border in LmActin also contains Gln263Lys substitution, and the lysine forms a charge pair with Glu258. This stabilizes the H-plug outwards from the center of the filament towards the pointed end tip. Furthermore, insertion of Asp269 to the H-plug introduces an additional buried negative charge to the core of the filament that

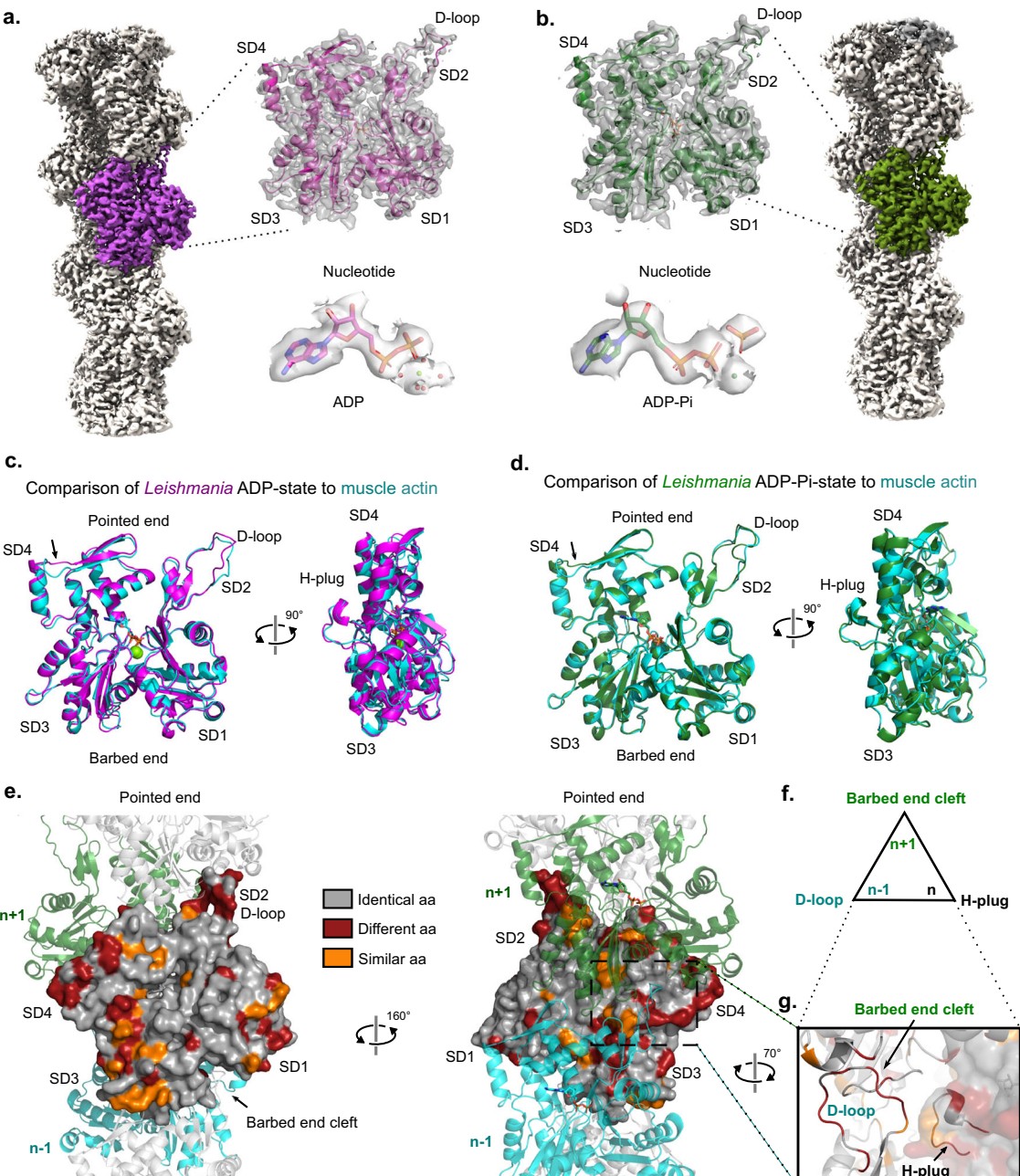

**Fig. 3 Structures of *Leishmania* ADP- and ADP-Pi-actin filaments. a** A cryo-EM map of *Leishmania major* ADP-actin filament. The overall model, ADP and waters modeled in the flattened density around the Mg$^{2+}$ ion are shown. **b** A cryo-EM map of *L. major* ADP-Pi-actin filament. The overall model, ADP, Pi, and Mg$^{2+}$ are shown. **c** Superimposition of *L. major* ADP-actin (magenta) to vertebrate ADP-actin (cyan, PDB ID: 6DJO) subunit. The pointed end tips of actin subdomain 4 (indicated with an arrow) follow slightly different paths in LmActin and vertebrate muscle actin. **d** Superimposition of *L. major* ADP-Pi-actin (green) to vertebrate ADP-Pi actin (cyan, PDB ID: 6DJN) subunits. The ADP-Pi conformations in *L. major* and vertebrate actins are nearly identical to each other. **e** Positions of residues that are not conserved between rabbit muscle and *L. major* actins are highlighted in the structure surface presentation. Gray color indicates identical, orange similar, and red different amino acids in the corresponding positions between the two actins (see Supplementary Fig. 3b). Neighboring subunits (*n*-1 and *n*+1) are shown in cyan and green cartoon representations. Note that the non-conserved residues are concentrated at the regions important for inter- and intra-strand contacts, with the exception of the barbed end groove that exhibits high conservation. **f** Schematic presentation of the regions that are critical for the formation of an actin filament core. **g** D-loop, together with barbed end groove and H-plug from two neighboring subunits form the core of the filament. Red and orange colors depict sequence conservation as described in panel **e**.

is not compensated by any additional charge pair from the neighboring subunits. Instead, the neighboring D-loop contains His40Asn and Arg39Lys substitutions. These alterations affect the electrostatic network formed between Glu270, Lys39, His173, and Asp286 (Fig. 4a). Overall, we speculate these alterations in LmActin weaken the central core of the filament that is formed by the three adjacent subunits.

At the pointed end tip of subdomain 4, we observed an interesting shift of the β-strand 237–242 towards the center of the filament when compared to muscle actin (Figs. 3c, d, 4b and

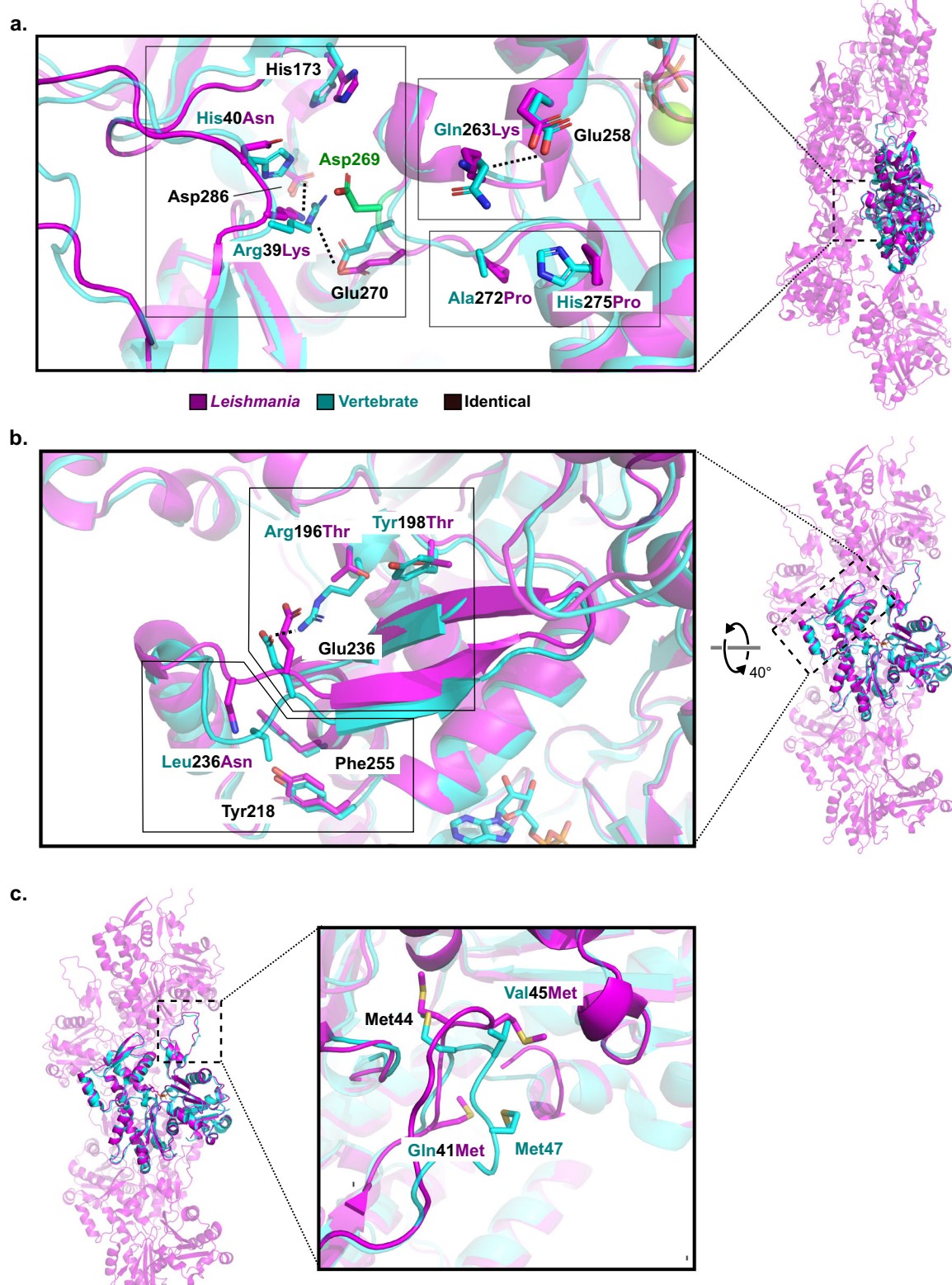

Supplementary Fig. 4). This is also close to the region where phalloidin and jasplakinolide toxins bind[40,49]. There are most likely two reasons for the movement. First, the connecting loop between α-helix 220–232 and β-strand 237–242 is shorter in LmActin due to a single amino acid deletion (Fig. 4b and Supplementary Fig. 3b). In muscle actin, this loop adopts a different conformation due to Leu236, which inserts in a small

hydrophobic pocket created by Tyr218 and Phe255. In LmActin, Leu236Asn substitution disrupts hydrophobic contacts and helical conformation of the loop. Second, two substitutions, Arg196Thr and Tyr198Thr, provide extra space for the movement of β-strand 237–242. The Arg196Thr substitution also breaks the electrostatic interaction with Glu236 (Fig. 4b). These differences suggest that the pointed end tip of LmActin is more

**Fig. 4 The differences in filament inter- and intra-strand interfaces between *Leishmania* and vertebrate actins. a** H-plug of LmActin (magenta) contains an insertion of an additional aspartate (Asp269, highlighted in green) in the core of the filament that is not present in vertebrate actin (cyan). Corresponding interaction partners at the neighboring D-loop also differ between the two actins. In LmActin, the electrostatic network, formed by Glu270, Lys39, and Asp286 to His173, is hindered by aspartate insertion. Ala272Pro and His275Pro substitutions, preceding the H-plug in LmActin, decrease the flexibility of the H-plug region in comparison with vertebrate actin. Lys263 (absent in the vertebrate actin) can form a salt bridge with Glu258 located in the pointed end helix of subdomain 4, and thus link the dynamics of the H-plug region to the pointed end of the monomer. **b** The pointed end face of LmActin subdomain 4 contains a single amino acid deletion leading to a shorter loop between α-helix 222–233 and β-strand 238–242. In vertebrate actin, this loop can adopt an α-helical conformation that stabilizes the upstream pointed end tip. In vertebrate actin, Leu236 is inserted into the hydrophobic pocket formed by Tyr218 and Phe255. In LmActin, Leu236Asn substitution and the amino acid deletion described above disrupt the hydrophobic contacts and the helical conformation of the loop. Furthermore, the Arg196Thr and Tyr198Thr substitutions in LmActin allow more room for the flexibility of the pointed end tip. **c** The D-loop of LmActin contains three methionines, from which the Met44 and Met45 insert deep into the adjacent barbed end groove of actin. In contrast, similar extensive hydrophobic insertion is not observed in vertebrate actin.

dynamic, and this may lead to weaker intra-strand interactions with the adjacent subunits, as also supported by the slightly lower quality of maps in this region (Supplementary Fig. 3a and Supplementary Fig. 4).

The amino acid composition of the D-loop of LmActin is very different from that of α- skeletal muscle actin (Supplementary Fig. 3b). Of particular interest are the three methionines in *Leishmania* actin, two of which (Met44 and Met45) are inserted deep into the barbed end cleft of the adjacent longitudinal subunit. The third methionine (Met41) is in close proximity to the C-terminus and H-plug of the adjacent actin subunits (Fig. 4c and Supplementary Fig. 4). We speculate that Met44 and Met45 could form more stable interactions with the adjacent subunit in comparison to vertebrate actins, and could thus compensate for weaker lateral subunit interactions within the filament. Another option is that the repetitive hydrophobic sequence in the tip of the D-loop has an impact on the overall dynamics of the D-loop. Earlier biochemical[50–52], and MD simulation[53] studies provided evidence that the D-loop samples various conformations in the filament. Some conformations may be more favorable for the association of the D-loop with the adjacent subunit, and these states can be affected by the D-loop primary sequence.

We additionally observed some amino acid differences in the nucleotide coordinating loops (Supplementary Fig. 3d). However, the residues around the proposed "backdoor" pathway for phosphate dissociation are conserved between the two actins. Thus, it is possible that both the dynamics of the nucleotide-binding pocket and subunit dynamics within the actin filament may contribute to the faster Pi-release from LmActin compared to vertebrate actins. Of note, the nucleotide coordinating loops are interconnected with the H-plug region, and together with the structural features explained above, can explain the faster phosphate release in LmActin.

To elucidate how the structural differences identified above may contribute to the rapid dynamics of LmActin filaments, we estimated by PDBePISA server the total free energy for dissociation of filament interfaces of different ADP-actin filament structures (Supplementary Table 2). Interestingly, LmActin exhibited the lowest free energy (−17.4 kcal/mol) when compared to chicken actin (−22.0 kcal/mol), jasplakinolide-stabilized PfActin (−21.2 kcal/mol) or phalloidin-stabilized rabbit actin (−24.3 kcal/mol). Taken together, the structural differences in the actin subunit interfaces of LmActin filaments described above are most likely responsible for their very rapid dynamics compared to muscle actin.

**Rapid severing of actin filaments by *Leishmania* cofilin.** ADF/cofilin severs ADP-actin filaments and is thus a central regulator of actin filament disassembly in all eukaryotes characterized so far[54]. All ADF/cofilin orthologs examined to date, however, do not sever actin filaments particularly efficiently. Thus, additional

cofilin "cofactors", including Aip1, are required for frequent actin filament severing and rapid disassembly in animal and yeast cells[55]. Interestingly, Aip1 orthologs have not been identified in *Leishmania* species, and thus how cofilin achieves rapid actin filament severing is not known[11]. We applied microfluidics-coupled TIRF microscopy to compare the severing frequencies of LmActin filaments by *L. major* cofilin (LmCofilin) and RbActin by mammalian cofilin-1. Experiments with mammalian cofilin-1, both unlabeled and mCherry-tagged, confirmed previously reported observations: cofilin-1 binds cooperatively to filaments and slowly severs them at domain edges (Fig. 5a, b and Supplementary Fig. 5). As a consequence, at high concentrations, cofilin-1 rapidly saturates filaments but barely fragments them (Supplementary Fig. 5)[56]. LmCofilin possessed a drastically different behavior: Filaments fragmented much faster than with mammalian proteins (100-fold at 500 nM cofilin), and we did not observe a decrease in the fragmentation rate at higher LmCofilin concentrations. These observations suggest that LmCofilin disassembles filaments in a different manner compared with mammalian cofilin-1. Filaments could fragment from within cofilin domains, or extremely efficiently at domain edges, before saturation by cofilin.

**Structural basis of actin filament severing by *Leishmania* cofilin.** To gain mechanistic insight into the remarkably efficient filament severing by *Leishmania* cofilin, we determined the atomic structure of the cofilin-bound LmActin filament. Interestingly, when we vitrified LmCofilin together with LmActin filaments for cryo-EM, the cofilin-actin filaments were typically more bent compared to predominantly straight bare actin filaments in the same views (Fig. 5c). 3D reconstruction of these filaments allowed us to determine the structure of the LmCofilin-decorated LmActin filament at an average resolution of 3.4 Å (Fig. 5d and Supplementary Fig. 3a). The LmCofilin-bound LmActin filament was slightly more twisted (−161.3°) when compared to other cofilin-actins characterized to date (−162.1 to 162.5°) (Supplementary Table 1)[57,58]. Also, the helical rise of the filament was slightly increased in comparison to the earlier structures (28.6 vs. 27.2–27.6 Å).

*Leishmania* cofilin interacts with the barbed end of an actin monomer between subdomains 1 and 3, similarly to mammalian cofilin and the mammalian ADF-H domains of twinfilin. Additionally, *Leishmania* cofilin interacts with the "F-site" located on the subdomain 1 and the subdomain 2 of the longitudinal adjacent actin subunit, similarly to mammalian cofilin[57,59,60] (Fig. 5e). The binding of LmCofilin to LmActin led to a similar rotation of the subdomains 1 and 2 relative to the subdomains 3 and 4 as described by earlier studies[57,61]. Thus, the overall conformational change in the actin subunit was nearly identical to the mammalian actin in complex with cofilin (Fig. 5f).

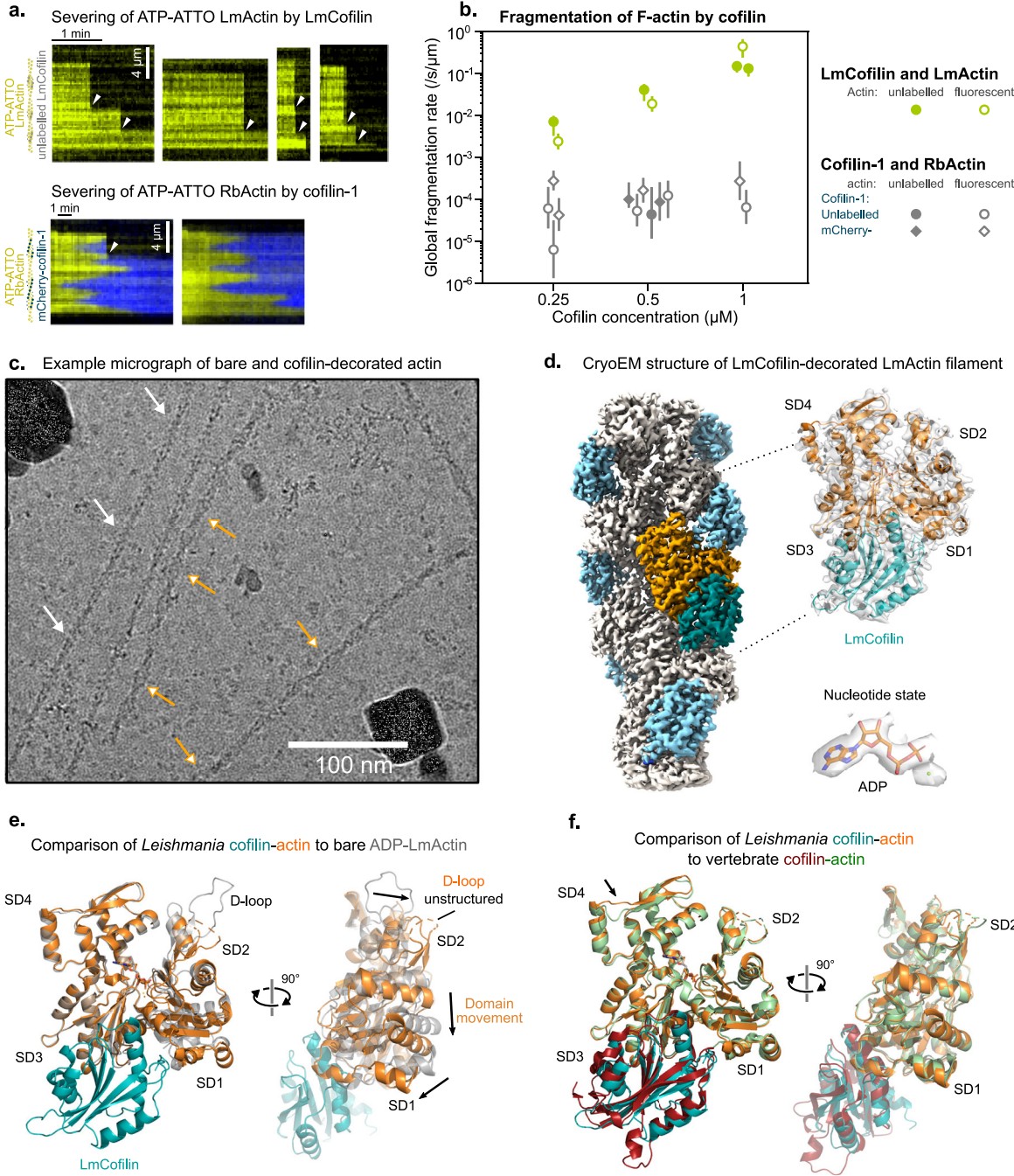

a. Severing of ATP-ATTO LmActin by LmCofilin

b. Fragmentation of F-actin by cofilin

c. Example micrograph of bare and cofilin-decorated actin

d. CryoEM structure of LmCofilin-decorated LmActin filament

Nucleotide state
ADP

e. Comparison of *Leishmania* cofilin-actin to bare ADP-LmActin

f. Comparison of *Leishmania* cofilin-actin to vertebrate cofilin-actin

Finally, as in the mammalian cofilin-actin complex, the D-loop of cofilin-bound LmActin becomes unstructured (Fig. 5e, f).

More careful analysis of the LmActin-cofilin structure revealed two interesting differences compared with the mammalian cofilin-actin (Fig. 6a–d). First, the N-terminal region of *Leishmania* cofilin harbors isoleucine in position 3 instead of serine 3, which can be phosphorylated in various cofilin orthologs. In the LmActin-cofilin structure, Ile3 forms specific contacts with the barbed end cleft of the actin subunit. When overlaying the D-loop position from the ADP-state of bare LmActin filament to the LmActin-cofilin structure, the hydrophobic Ile3 collides with the position of Met45 in the D-loop. The hydrophobic N-terminus of cofilin can thus potentially displace the hydrophobic tip (Met44 and Met45) of the D-loop from its position at the barbed end cleft (Fig. 6d). Second, the C-terminus of the *Leishmania* cofilin differs from that of mammalian cofilin. In mammalian cofilins, the C-terminal region harbors two additional β-strands, which associate with the neighboring two β-strands of cofilin helping to orient the loop 91–96 for actin-binding. In LmCofilin, the two β-strands are absent from the C-terminal region, but the C-terminus instead forms an extended α-helix, which comes to close proximity with the nucleotide-binding cleft between actin subdomains 2 and 4. This resembles a "battering ram" that pushes the subdomain 2 towards a non-flattened conformation, reminiscent of the monomeric conformation (Fig. 6a, b), and forms a specific charge pair, hydrogen bond, and hydrophobic contacts with the actin surface (Fig. 6c). Of note, also yeast cofilin has an extended C-terminus, and its overall structure is quite similar when compared to LmCofilin (Supplementary Fig. 6a).

**Fig. 5 Rapid severing of *Leishmania* actin filaments by cofilin. a** Typical kymographs showing the fragmentation of ATP-ATTO-labeled ADP-actin filaments by 500 nM cofilin, comparison between *Leishmania* and vertebrate proteins. White arrowheads indicate severing events. Note the difference in time scales, and that severed fragments disappear from the kymographs due to flow in the microchamber. **b** Comparison of the global fragmentation rate of LmActin by LmCofilin (green) and RbActin by vertebrate cofilin-1 (gray). Each data point was calculated by fitting the survival fraction over different F-actin segments (*n*) with a single exponential and normalized by the mean segment length (units: /s/μm, see "Methods"). Error bars represent 95% confidence intervals. For unlabeled LmActin: 0.25 μM $N = 1$, $n = 90$; 0.5 μM $N = 1$, $n = 90$; 1 μM $N = 2$, $n = 90,90$. For fluorescent LmActin: 0.25 μM $N = 1$, $n = 20$; 0.5 μM $N = 1$, $n = 20$; 1 μM $N = 1$, $n = 20$. For unlabeled RbActin with unlabeled cofilin-1: 0.5 μM $N = 1$, $n = 50$. For fluorescent RbActin with unlabeled cofilin-1: 0.25 μM $N = 2$, $n = 30$; 0.5 μM $N = 2$, $n = 30$; 1 μM $N = 1$, $n = 30$. For unlabeled RbActin with mCherry-cofilin-1: 0.5 μM $N = 2$, $n = 22, 40$. For fluorescent RbActin with mCherry-cofilin-1: 0.25 μM $N = 2$, $n = 20, 50$; 0.5 μM $N = 1$, $n = 50$; 1 μM $N = 1$, $n = 50$. **c** Example micrograph (dose-weighted and drift-corrected, low-pass filtered to 20 Å) from a cryo-EM sample containing LmActin actin and LmCofilin. Orange arrows with white heads indicate cofilin-decorated filaments, which were often curved. White arrows indicate bare LmActin filaments, which were predominantly more straight compared to cofilin-actin filaments. **d** The cryo-EM map of LmActin filament decorated with *Leishmania* cofilin. Density for the model, around ADP and associated Mg$^{2+}$ ion are shown. **e** Comparison of the actin subunit conformations of cofilin-bound (orange) and bare (gray) LmActin filaments superimposed on subdomains 3 and 4. Flat conformation of the LmActin subunit undergoes a propeller-like conformational change from subdomains 1 and 2 when bound to cofilin. The D-loop in the cofilin-bound filament is unstructured. **f** Comparison of *Leishmania* cofilin-actin (in cyan and orange) to vertebrate cofilin-actin (in red and green, from PDB ID: 5YU8). The actin structures represent similar conformations, but the subdomain 4 at the pointed end tip (black arrow) follows slightly different tracks between the two cofilin-bound actins.

---

Next, we tested the contribution of the C-terminus of *Leishmania* cofilin on actin filament severing and generated a mutant *Leishmania* cofilin lacking the four C-terminal residues (D4C). Compared to the wild-type *Leishmania* cofilin, the C-terminally deleted protein severed LmActin filaments far less efficiently. At a higher concentration (5 μM), this mutant exhibited moderate severing activity, suggesting that the C-terminus of *Leishmania* cofilin is critical for both high-affinity binding and severing of LmActin filaments (Fig. 6e). Interestingly, although the main cofilin-binding interface is relatively well-conserved between *Leishmania* and mammalian actins (with the exception of the D-loop sequence, Supplementary Fig. 6b), the labeled mouse cofilin-1 failed to bind LmActin filaments in our experiments (Fig. 6f). These data suggest the D-loop sequence composition may have an important role in determining cofilin binding. This is in line with a study showing that D-loop point-mutations altered cofilin binding[51]. Interestingly, also LmCofilin did not fragment RbActin filaments (Supplementary Fig. 6c) indicating the two cofilins have diverged to bind only their corresponding actin sequence.

Together, these experiments reveal that *Leishmania* cofilin severs LmActin filaments remarkably rapidly. Although the overall conformations of *Leishmania* and mammalian cofilin-actin filaments are very similar to each other, there are key differences in the interactions of the N- and C-termini of cofilins with actin filaments. These, and the slightly more twisted nature of the filaments, are likely to account for the differences between *Leishmania* and mammalian cofilins in actin filament binding and severing.

***Leishmania* twinfilin regulates actin filament barbed end dynamics**. Two conserved actin-binding proteins, Capping protein and twinfilin, together participate in the dynamics of actin filament barbed ends in mammalian and yeast cells[16,60,62,63]. However, Capping protein is not present in *Leishmania* species[11]. Moreover, although *Leishmania* express a homolog of twinfilin, a previous study reported that it does not bind actin[21]. To elucidate how actin filament assembly and disassembly are controlled at filament barbed ends in *Leishmania*, we expressed and purified *L. major* twinfilin (LmTwf) and examined its possible effects on actin dynamics. We tested the activity of LmTwf by assembling "segmented" filaments exposed sequentially to solutions containing 1 μM unlabeled LmActin and 1 μM LmProfilin, with or without LmTwf. We found that in the presence of assembly-competent actin monomers LmTwf drastically reduces the filament barbed end polymerization rate and even triggers

depolymerization at concentrations higher than 0.2 μM, down to ~8 subunits/s. (Fig. 7a). This result cannot be explained solely by LmActin sequestration by LmTwf, but suggests that LmTwf interacts with the filament barbed end and prevents the addition of actin monomers. Moreover, the depolymerization rate of ADP-LmActin filaments in the absence of actin monomers is reduced when exposed to LmTwf (to ~3 subunits/s), compared with buffer only (Fig. 7b). These two observations are in line with the recent results obtained on mammalian actin and twinfilin[16,63]. Interestingly, LmTwf also slows down the depolymerization of ADP-RbActin filaments, and even more potently than what was reported for mammalian twinfilin (to ~0.5 subunits/s vs. 5.0 subunits/s[16]) (Supplementary Fig. 7). Together, these experiments reveal that *Leishmania* twinfilin is an actin-binding protein, which efficiently inhibits actin filament barbed end polymerization, while simultaneously allowing depolymerization with a rate up to ~8 subunits/s.

## Discussion

Studies from the past decades have led to a considerably good understanding of the molecular mechanism controlling actin dynamics and actin-dependent cellular processes in animals, yeasts, and plants[45]. However, our knowledge of the actin cytoskeletons of more divergent organisms is limited, and thus the evolution and robustness of the actin turnover machineries are incompletely understood. Here, we determined the structure of actin filament from a flagellated parasite, *L. major*, representing the most divergent actin filament structure reported so far. This reveals interesting similarities and differences between metazoan and flagellated parasite actins. Moreover, we demonstrate that *L. major* actin filaments are more dynamic compared to extensively-studied mammalian actins, and that *Leishmania* cofilin is a very effective disassembler of *Leishmania* actin filaments.

We reveal that LmActin assembles into single, dynamic filaments, which polymerize readily under typical conditions for canonical actins. This is in contrast to an earlier study, in which a homologous His-tagged *L. donovani* actin displayed peculiar polymerization properties and predominantly assembled into filament bundles[10]. These differences could be explained by the presence of His-tag in the actin used in this earlier study. Importantly, our microfluidics-microscopy experiments revealed that LmActin filaments are significantly more dynamic compared with mammalian actin filaments. Barbed and pointed end depolymerization rates of *Leishmania* ADP-actin were ~5-fold and ~20-fold more rapid, respectively, when compared with the corresponding rates of mammalian α-skeletal muscle actin and

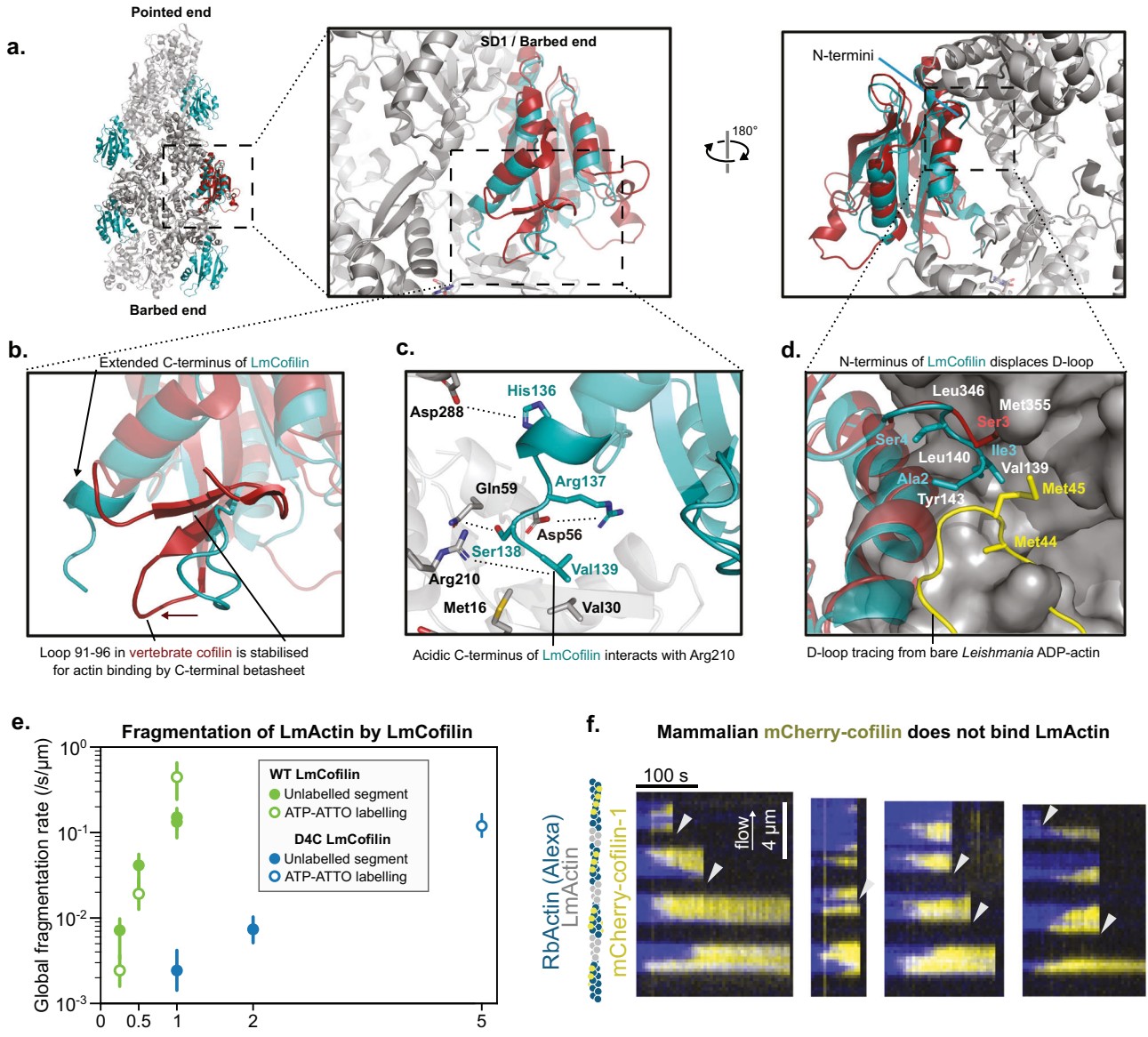

**Fig. 6 Structural basis of *Leishmania* actin filament severing. a** Structure of the LmActin filament decorated with *Leishmania* cofilin. In comparison to vertebrate cofilin (in red), the *Leishmania* cofilin (in cyan) contains an extended C-terminal α-helix. **b** A close view of the C-terminal region of cofilin. In vertebrate cofilin, C-terminus forms a turn followed by a two-strand β-sheet, which stabilizes loop 91–96 to closer contact with the actin filament. This β-sheet is absent from *Leishmania* cofilin, which contains an extended C-terminal α-helix. **c** Amino acid contacts between the C-terminus of *Leishmania* cofilin and LmActin. The acidic C-terminus can form an ion pair with Arg210 in actin, and local hydrophobic contacts through Val139 to Met16 and Val30. His136 and Arg137 can form ion bonds to Asp288 and Asp56, respectively, in actin. **d** The N-terminus of *Leishmania* cofilin interacts with the barbed end of actin. In the cofilin-bound actin, the D-loop is unstructured (not shown in the figure), whereas in bare actin filaments the D-loop (shown in yellow) associates with the hydrophobic pocket (residues named in white) of the adjacent actin subunit. Hydrophobic Ile2 from *Leishmania* cofilin collides with Met45 of D-loop, and is thus expected to replace the D-loop from the hydrophobic pocket. In vertebrate cofilin (cyan), the N-terminal tracing of the structure terminates to Ser3. **e** Global severing rates of wild-type *Leishmania* cofilin (data from Fig. 5b) and a mutant *Leishmania* cofilin lacking the four C-terminal amino acids (D4C). Each data point was calculated by fitting the survival fraction of >60 F-actin segments (*n*) with a single exponential and normalized by the mean segment length (units: /s/μm, see Methods). Error bars were calculated by fitting the upper and lower values of the 95% confidence interval. For unlabeled segments with wild-type LmCofilin: 0.25 μM *N* = 1, *n* = 90; 0.5 μM *N* = 1, *n* = 90; 1 μM *N* = 2, *n* = 90, 90. For ATP-ATTO labeled segments with wild-type LmCofilin: 0.25 μM *N* = 1, *n* = 20; 0.5 μM *N* = 1, *n* = 20; 1 μM, *N* = 1, *n* = 20. For unlabeled segments with D4C LmCofilin: 1 μM *N* = 1, *n* = 90; 2 μM *N* = 1, *n* = 240. For ATP-ATTO labeled segments with D4C LmCofilin: 5 μM *N* = 1, *n* = 60. **f** Decoration and fragmentation of segmented filaments (Alexa-RbActin segments in blue, and unlabeled LmActin segments) by 200 nM vertebrate mCherry-cofilin-1 (yellow) from a single experiment. No decoration by mCherry-cofilin-1 is observed on unlabeled LmActin segments. The severing of filaments appears to occur mainly at the interface between LmActin and rabbit actin segments (arrowheads).

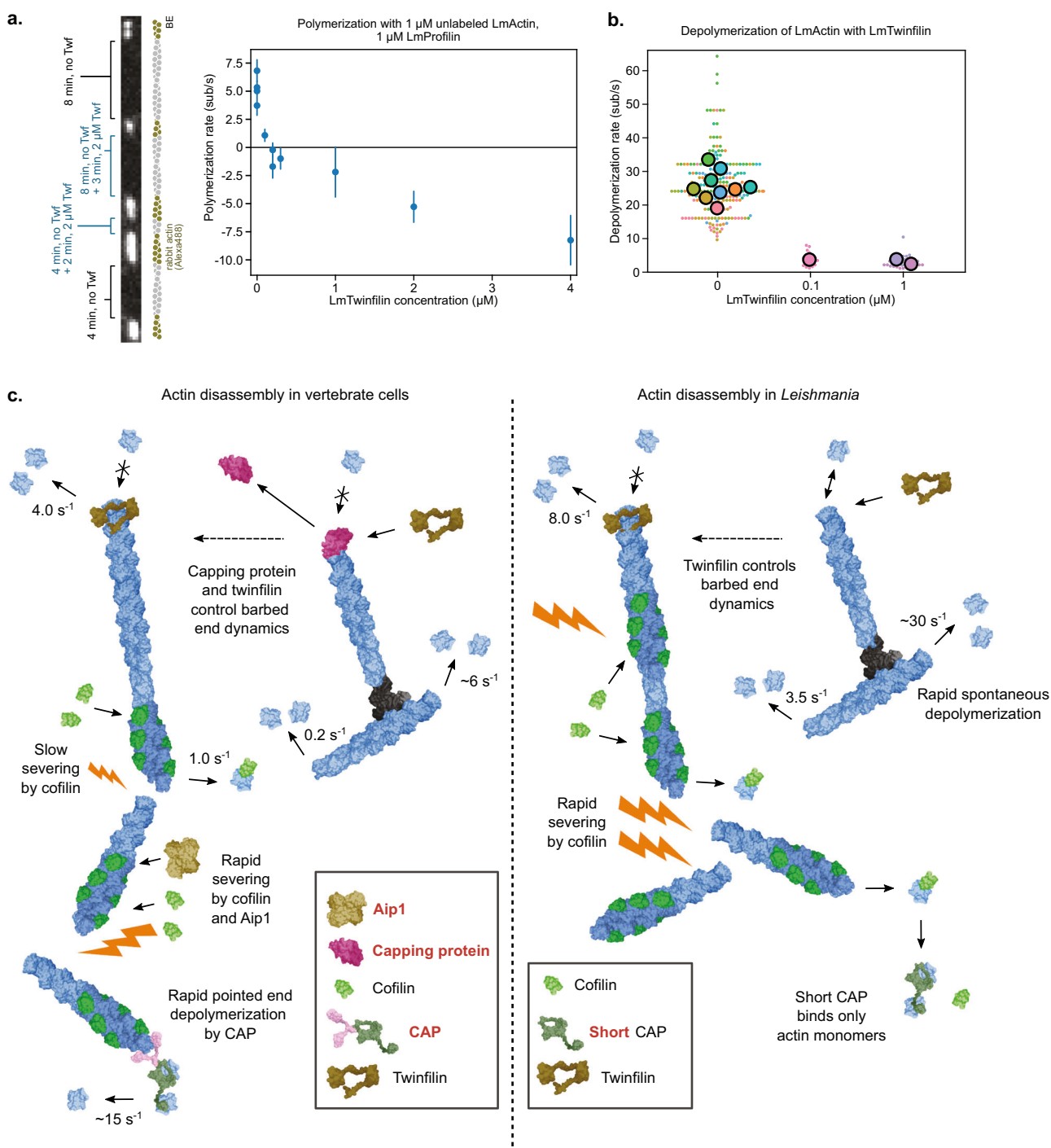

**12**        NATURE COMMUNICATIONS | (2022)13:3442 | https://doi.org/10.1038/s41467-022-31068-y | www.nature.com/naturecommunications

cytoplasmic actin. Inorganic phosphate release from LmActin filaments was ~5–10 times faster than from mammalian α-skeletal muscle actin. Earlier studies have indicated that malaria parasite actin (PfActin1) also appears to display very different biochemical properties compared to canonical actins, because it forms only very transient filaments. However, the precise dynamics of PfActin1 filaments is still a matter of controversy due to different results obtained with different methodologies[37,39]. Of note, *L. major* and malaria parasite actins are highly divergent from each other, LmActin displaying only ~70% sequence identity to both mammalian and malaria parasite actins. Moreover, actin in malaria parasites has a very specific function, gliding motility, which likely requires unique dynamic properties. On the other hand, in trypanosomatids, such as *Leishmania*, actin

appears to be required in endocytosis similarly to yeasts and animal cells[11].

Despite their different dynamics compared to other actins characterized so far, *L. major* actin filaments are structurally remarkably similar to actin filaments from other species[40,47,48,64–70]. Our experiments demonstrate that actins even with large evolutionary distances can co-polymerize with each other, although the subunit interfaces are not optimal in the heterologously-formed actin filaments. This translates into more rapid barbed end depolymerization rates, and conversely, into pointed ends being 'capped' by muscle actin exhibiting slower dynamics.

There are however key structural differences between *Leishmania* and vertebrate actins that may provide an explanation for

**Fig. 7 Principles of rapid actin disassembly in *Leishmania* parasites. a, b** Impact of *Leishmania* Twinfilin (LmTwf) on LmActin barbed end dynamics. (panel **a**) Barbed end polymerization rate in the presence of 1 μM LmActin, 1 μM *Leishmania* profilin, and indicated concentrations of LmTwf. Negative values indicate that there is net depolymerization. Each point represents the mean over different actin segments (*n*) from independent experiments (*N*). Error bars represent S.D. $N = 4$, $n = 20, 20, 40, 40$ for 0 μM; $N = 1$, $n = 20$ for 0.1 μM; $N = 2$, $n = 40,40$ for 0.2 μM; $N = 1$, $n = 40$ for 0.3 μM; $N = 1$, $n = 40$ for 1 μM; $N = 1$, $n = 40$ for 2 μM; $N = 1$, $n = 40$ for 4 μM. (panel **b**) Depolymerization rate of unlabeled ADP-LmActin exposed to F-buffer, with and without LmTwf. Large symbols represent averages over individual measures (*n*) (small symbols) from independent experiments (*N*). $N = 9$, $n = 30, 32, 32, 19, 40, 20, 20, 20, 20$ for 0 μM; $N = 1$, $n = 20$ for 0.1 μM; $N = 2$, $n = 15, 20$ for 1 μM. **c** Model for the rapid actin disassembly in vertebrate and *Leishmania* cells. In mammalian cells, Capping protein (CP) and twinfilin cooperate to control actin assembly at filament barbed ends. CP is not expressed in *Leishmania*, but mammalian and *Leishmania* twinfilin can both prevent filament barbed end polymerization and allow filament barbed end depolymerization. Mammalian ADF/cofilins sever actin filaments quite inefficiently, and the severing frequency is enhanced by Aip1. This cofilin 'cofactor' is absent from *Leishmania*, but *Leishmania* cofilin severs LmActin filaments much more efficiently compared to mammalian proteins. Filament pointed end depolymerization of mammalian actins is relatively slow both for bare and cofilin-actin filaments, but the depolymerization of cofilin-actin filaments is greatly accelerated by cyclase-associated protein (CAP). *Leishmania* 'mini-CAP' lacks the N-terminal protein domain that catalyzes actin filament depolymerization. However, LmActin filaments depolymerize at their pointed ends ~20-fold more rapidly compared to mammalian actin filaments. The pointed end depolymerization rates of *Leishmania* cofilin-actin filaments could not be measured due to rapid severing. Please note that the model presents each reaction only in one direction, which is considered physiologically most relevant for actin filament disassembly, while reverse reactions also occur in each case.

their biochemical differences. Most notably we observed structural differences between these actins in the hydrophobic core consisting of the D-loop, H-plug, and barbed end cleft, which are critical in defining filament stability[71–73]. However, the precise roles of different elements in this region for actin filament dynamics are still incompletely understood. For example, it is still unclear whether the population of D-loop conformations (open/closed) is related to the stability of actin filament[47–49,70], and how the primary sequence defines the stability and dynamics of the D-loop—barbed end interaction. The conformational dynamics of the D-loop could indirectly affect the filament core stability through interactions with H-plug and vice versa. An MD simulation study showed that the D-loop is important in stabilizing the pointed end of actin filaments through lateral interactions with the H-plug of the adjacent subunit[53]. Furthermore, ref. [74] achieved stabilization of Apicomplexan actin by mutating the H-plug and the actin subunit pointed end, whereas others[39,75] were able to stabilize malaria actin filaments by switching its D-loop sequence to the one from muscle actin. In this context, it is important to note that although the overall conformations of *Leishmania* and malaria parasite actins are similar to each other (Supplementary Fig. 3c), the key contacts in the H-plug, D-loop, and the actin subunit pointed end regions are different (Supplementary Fig. 8). It is also worth noting that the malaria actin filament structures were determined in the presence of the actin filament stabilizing drug, jasplakinolide, which could affect the local conformation of this region[40,49]. Nevertheless, the specific structural features of the hydrophobic core are critical for providing different dynamic properties for the animal, Apicomplexan and flagellated parasite actin filaments.

We revealed that *Leishmania* cofilin severs LmActin filaments much more frequently compared with mammalian proteins. Although the overall structure of *Leishmania* cofilin-actin filament is very similar to the corresponding structure of vertebrate proteins, there are some interesting differences. First, the C-terminal α-helix of *Leishmania* cofilin is extended compared to the one from vertebrate cofilins. Our mutagenesis studies also demonstrate that the extended α-helix is important for efficient actin filament severing by *Leishmania* cofilin. Interestingly, budding yeast cofilin also contains an extended C-terminal α-helix, similar to that of *Leishmania* cofilin, but its possible role in actin filament binding and severing has not been reported. At least in *C. elegans* UNC-60B cofilin, the C-terminus seems to play a critical role in actin filament binding and severing[76]. The second notable difference between *Leishmania* and mammalian cofilins is in the N-terminus, which in *Leishmania* harbors isoleucine instead of serine in position 3. This isoleucine makes

contact with the hydrophobic patch in the barbed end cleft, and may thus be responsible for displacing the D-loop of an adjacent subunit from this site in the cofilin-actin structure. This hypothesis is supported by a recent study on mammalian proteins demonstrating that the 'phosphomimetic' Ser3Asp mutant cofilin still binds to the actin filament, but is unable to induce a conformational change in the filament. Moreover, this earlier study demonstrated that in the presence of Ser3Asp mutant cofilin, the D-loop is still associated with the hydrophobic patch in the barbed end of the adjacent subunit[58].

Mammalian ADF/cofilins sever actin filaments at the interface between cofilin-decorated and bare filaments[56,77–79]. Despite extensive efforts, we did not succeed in preparing a functional labeled *Leishmania* cofilin. An identical N-terminally tagged mCherry construct to mammalian cofilin failed to bind actin filaments highlighting the importance of the "native" N-terminus in *Leishmania* cofilin. Future studies are needed to uncover whether *Leishmania* cofilin also severs at the interface of bare and cofilin-decorated actin segments, or if the rapid severing by *Leishmania* cofilin could result from its ability to also destabilize the actin filaments within the cofilin-decorated segments. Moreover, mammalian ADF/cofilins were shown to promote filament depolymerization at both ends[12], and whether these activities are conserved in *Leishmania* cofilin remains to be determined.

Similar to yeasts and animal cells, actin is associated with endocytosis in trypanosomatids[7]. These parasites also express the Arp2/3 complex, which nucleates the branched, dynamic actin filament network that provides force for endocytosis[80]. Our biochemical data provide a plausible explanation for how rapid actin filament turnover can be achieved in *Leishmania* parasites in the absence of Aip1, which accelerates filament severing, and in the presence of a short version of the cyclase-associated protein, which lacks the protein domain that is critical for accelerating actin filament pointed end depolymerization in 'higher' eukaryotes (Fig. 7c). *Leishmania* cofilin severs actin filaments very efficiently, and this may bypass the need for Aip1 in this organism. Moreover, bare *Leishmania* actin filaments depolymerize from their pointed ends ~20 times more rapidly compared with bare α-skeletal mammalian muscle and cytoplasmic actin filaments. Due to the very rapid filament severing, we could not reliably measure the pointed end depolymerization rates of *Leishmania* cofilin-actin filaments. However, assuming that *Leishmania* cofilin accelerates filament pointed end depolymerization to a similar extent as mammalian ADF/cofilins[12], *Leishmania* actin filaments would depolymerize from their pointed end as fast as mammalian actin filaments in the presence of both cofilin and cyclase-associated protein[30,31]. *Leishmania* also lacks

Capping protein, which together with twinfilin controls actin filament barbed end dynamics in yeasts and animals[16]. Because *Leishmania* twinfilin can both prevent actin filament assembly and simultaneously allow filament disassembly at the barbed end, we envision that filament barbed end dynamics can be controlled in *Leishmania* by twinfilin alone. Thus, it is possible that Capping protein has a more complex role related to filament nucleation and assembly as also evidenced by recent studies[24,81,82].

What could be the reason for a limited number of proteins controlling actin dynamics in *Leishmania*? We speculate that *Leishmania* parasites have only a very simple actin cytoskeleton, which is predominantly involved in endocytosis. Inherently dynamic actin filaments are well-suited for endocytosis in *Leishmania* parasites, because endocytosis requires rapid actin filament turnover[83]. On the other hand, more complex regulation of the actin cytoskeleton by additional proteins (e.g., Aip1, Capping protein, filament depolymerizing CAP, tropomyosins) in "higher" eukaryotes may be linked to multiple cellular functions of the actin cytoskeleton in these organisms. These different, co-existing cellular functions require several different actin filament populations with specific dynamics, protein compositions, and sub-cellular localization. We speculate that the ancient actin filaments were inherently dynamic. Coinciding with the increasing complexity of the actin cytoskeleton and the number of actin-dependent cellular functions in "higher" eukaryotes, the actin filaments may have evolved to become more stable. The simultaneous appearance of additional proteins promoting actin filament disassembly enabled to fine-tune the dynamics of different co-existing actin filament populations.

Collectively, this study demonstrates how rapid actin dynamics can be achieved through a small number of actin-regulatory proteins in flagellated parasites, and thus elucidates the evolution of the actin-regulatory machinery. By determining the structures of *L. major* ADP-, ADP-Pi-, and cofilin-actin filaments, we also provide plausible structural explanations for the specific biochemical properties of *Leishmania* actin filaments. These actin structures, together with the biochemical analysis of *Leishmania* actin, lay a foundation for future systematic mutagenesis and MD simulation studies, as well as open new avenues for developing specific inhibitors against trypanosomatids actins. Many *Leishmania* and related *Trypanosoma* species are devastating pathogens for humans and cattle, and there is a need of developing specific drugs against these parasites. In the future, it will be also interesting to reveal the structural and biochemical similarities and differences between *Leishmania* actin and actin-binding proteins compared to even more distant actins from Asgard archaea and *Giardia* parasites[84–86]. Such studies would further expand our understanding of the different ways by which desired actin dynamics can be achieved in evolutionarily divergent organisms.

## Methods

**Phylogenetic tree and sequence alignments**. The phylogenetic tree was generated in Geneious Prime (Build 2020-01-14). Alignment was based on the blosum45 score matrix and genetic distance model from Jukes-Cantor. Protein sequences were obtained from Uniprot, and the accession codes are provided in the Source Data file. To compare sequence identity and similarity between rabbit and *Leishmania* actins, sequences were pairwise aligned in Geneious Prime (Build 2020-01-14) with the "Clustal Omega" function using blosum45 score matrix with a threshold 2.

**Cloning**. We synthesized a codon-optimized actin gene (Thermo Scientific) of *Leishmania major* (LmActin) for insect cell expression based on reported Uniprot sequence Q9U1E8, corresponding to TriTrypDB:LmjF.04.1230 entry in TriTrypDB. Another entry, P45520, reported in 1995 is 100% similar and 99% identical (disagreements Q9U1E8 [93]EL[94], P45520 [93]DV[94]). For this study, we selected the more recently reported sequence from the year 2000. To maintain actin monomeric for the enrichment steps, we fused a chymotrypsin-cleavable human β-thymosin to the C-terminus of the protein with a 14-residue linker sequence and C-terminal His-tag (ASRGGSGGSSGGSASDKPDMAEIEKFDKSKLKKTETQEK NPLPSKETIEQEKQAGESHHHHHHHHHHHH), based on ref. [41]. The synthetic fragment containing a linker, the human-β-thymosin gene, and the Hisx10-tag was cloned to the C-terminus of the actin gene in pFastBac-1 (Thermo Scientific) baculovirus-delivery vector by NEBuilder HiFi DNA Assembly Master Mix (NEB, #E2621) following a standard manufacture protocol.

All *Leishmania major* genes (except for the actin gene) were codon-optimized for bacterial expression using the free online IDT Codon Optimization Tool, which is based on the frequencies of each codon's usage in the target organism, also known as the codon sampling strategy. The gene fragments were synthesized at TWIST Biosciences (San Francisco, USA), and then PCR-amplified and the overlapping regions of varying lengths were introduced on both ends of the fragments for further assembly reactions. The vectors were also PCR-linearized. All PCR reactions were carried out with KAPA HiFi HotStart ReadyMix (Roche, #7958927001), following the standard manufacturer protocol. Linear vectors were assembled with linear synthetic fragments with overlaps using NEBuilder HiFi DNA Assembly Master Mix (NEB, #E2621) following standard manufacture protocol. All final constructs were confirmed by Sanger-sequencing, and the primers and database references used to generate the constructs are described in Supplementary Table 3.

**Generation of baculovirus strain**. pFastBac-1-LmActin vector (Thermo Scientific) was transformed to MAX Efficiency DH10Bac (Thermo Scientific) to recombinantly generate the bMON14272 shuttle vector for insect cell transfection. Bacmid was isolated with Maxiprep (Macherey-Nagient). Subsequently, low-passage ExpiSF™ cells (Thermo Scientific, A35243) were transfected according to the manufacturer's protocol on a 25 mL scale to produce a P0 stock of baculovirus. P0 virus was aliquoted and frozen in liquid N$_2$ for long-term storage. To generate a sufficient amount of virus for large-scale expression, the P1 virus was generated according to the manufacturer's protocol. Briefly, 0.5–1 mL of P0 virus was used to infect a 400–800 mL culture at $2 \times 10^6$ cell density with the exception of the virus being collected by snap-freezing the culture in 10 mL aliquots after ~72 h of infection and stored in liquid nitrogen. Each P1 virus was titrated for optimal expression and growth of the cells.

**Expression and purification of *Leishmania* actin**. About $6 \times 1000$ mL of ExpiSF9™ (Thermo Scientific, A35243) cells were seeded at $5 \times 10^6$ cell density in 3 L bottles a 16–24 h before infection with the P1 virus. ExpiSf enhancer was added at 3.2 mL/ liter of culture. After 16–24 h cells were transfected with freshly thawed P1 virus. Infected cells were collected after 3–4 days of expression by pelleting at $1000 \times g$ with a J6-MI centrifuge for 25 min at +4 °C. Pelleted cells were washed with ice-cold phosphate-buffered saline (PBS) and pelleted again in 50 mL falcon tubes at $1000 \times g$ for 20 min. Pellets were snap-frozen and stored at −80 °C or immediately used for actin purification on the same day. For lysis of the cells, pellets were first suspended into a volume of ~10 times pellet volume in lysis buffer (20 mM HEPES, 300 mM NaCl, 10 mM imidazole, 0.5 mM ATP, 0.5 mM DTT, 0.25 mM CaCl$_2$, pH 7.4) containing Roche Complete ULTRA EDTA-free protease inhibitor cocktail and Pierce Universal Nuclease for cell lysis. Cells were first carefully solubilized by magnetic stirring and then lysed by passaging the solution through an EmulsiFlex C-3 homogenizer (Avastin) at 10,000–15,000 psi three times. After homogenization, the lysate was clarified by centrifugation at $38,759 \times g$ for 60 min. The supernatant was subsequently filtered with 0.45 μm nitrocellulose membrane (Millipore) and then loaded to a 5 mL HisTrap™ FF Ni-NTA column (GE Healthcare) equilibrated with lysis buffer and coupled to an ÄKTA™ Pure chromatography system (Cytiva). The column was washed with 30 x column volume (CV) of lysis buffer. The second wash step was performed with a linear gradient of 15 x CV reaching a final concentration of 15% elution buffer (lysis buffer containing 250 mM imidazole) and continued at 15% concentration for 5 x CV. Proteins were eluted with 10 x CV of 100% of elution with the reversed flow, yielding a sharp elution peak of ~4–6 mL, which was subsequently loaded to a HiLoad 16/60 Superdex 200 gel filtration column (GE Healthcare) equilibrated in PBS (containing 0.2 mM ATP, 0.2 mM CaCl$_2$ and 1 mM DTT). The protein was eluted at a volume of ~82 mL corresponding to a molecular size of ~50 kDa (based on gel filtration standards shown in ref. [30]). The peak was then concentrated with Amicon Ultra-4 30 kDa centrifugal filter (Merck), and snap-frozen in small aliquots for long-term storage. All steps until this point were executed on ice or at +4 °C. To remove the β-thymosin-10xHis fusion tag for subsequent experiments, frozen aliquots of LmActin from gel filtration were thawed and suspended in gel filtration buffer at a final concentration of ~0.3 mg/ml. α-chymotrypsin (Sigma-Aldrich, C3142) was then added in ~1:300–1:800 (w/w) to the actin solution and cleavage was performed ~16 h on ice. EGTA and MgCl$_2$ were added to final concentrations of 0.4 and 1 mM, respectively, and incubated for 1–2 h at room temperature. The cleavage reaction was quenched with phenylmethylsulfonyl fluoride or 4-(2-aminoethyl)benzenesulfonyl fluoride hydrochloride. Actin was centrifuged at $124,759 \times g$ using a TLA120.1 ultracentrifugal rotor at +10 °C for 1 h. The supernatant was discarded, and the visible pellet was washed 2–3 times with 1 mL of ice-cold G-buffer (5 mM Tris·HCl pH 7.4, 0.2 mM CaCl$_2$, 0.2 mM ATP, 0.5 mM β-mercaptoethanol). Finally, the pellet was rigorously resuspended to a final concentration of ~0.8 mg/ml for dialysis against G-buffer (at least

$3 \times 330$ mL). Before performing experiments, LmActin was spinned for 60 min at $124,759 \times g$ in TLA120.1 at $+10\,°C$. We note that despite polymerization/depolymerization cycles, some contaminating bands were present in the actin prep (Supplementary Fig. 1a) and these appeared during the cleavage. They are thus most likely products of unspecific cleavage. Electron spray ionization mass spectrometry confirmed the molecular weight of the majority of the product to be ~41,940 daltons. This roughly corresponds to the LmActin sequence with the first methionine removed and N-terminus acetylated, and for single methylation (most likely His73), with a theoretical calculated molecular weight of ~41,944 Da.

**Expression and purification of LmCofilin, LmProfilin, and LmTwinfilin.**
Recombinant LmCofilin, LmTwinfilin, and LmProfilin were expressed at $+22\,°C$ in *E.coli* BL21(DE3) (Merck Millipore) cells by using LB autoinduction media (AIMLB0210, Formedium) for ~24 h. Proteins were purified with a similar workflow as described in ref. [30]. First, His-tagged proteins were enriched by Ni-NTA column (GE HealthCare) and His-SUMO or His-GST removed by cleavage with SENP2 enzyme or 3 C protease, respectively. The purification tags were captured by treatment with $Ni^{2+}$-beads before proteins were gel filtered with Superdex 75 column (Cytiva) equilibrated in 20 mM HEPES, 50 mM NaCl, and pH 8 buffer. Proteins were concentrated with Amicon Ultracentrifugal filters (Merck) and snap-frozen with liquid nitrogen for long-term storage at $-75\,°C$.

**Microfluidic experiments.** Experiments were performed in Poly Dimethyl Siloxane (PDMS, Sylgard) chamber, typically 20 μm in height, 800 μm in width, and 1 cm in length, shaped into a cross with three inlets and one outlet. Chambers were mounted onto glass coverslips previously cleaned in successive ultrasound baths containing 2% Hellmanex (30 min, 35 °C), and 2 M NaOH (20 min, RT). Coverslips were thoroughly rinsed with milli-Q water after each bath, and stored in 100% absolute ethanol for up to 2 weeks. PDMS chambers and glass coverslips surfaces were exposed to UV, bound together, and left on a hot plate (100 °C, 3 min) to create a permanent bound. Controlled pressures were applied to the tubes containing the protein solutions, and the resulting flow rate was measured using microfluidic devices MFCS-EZ and Flow Units, respectively (Fluigent). To prepare the microfluidic chamber for experiments, solutions were first directly injected through a pipet tip plugged into the chamber outlet. Chambers were first passivated with PLL-PEG (1 mg/mL, 1–12 h, SuSos), functionalized with spectrin-actin seeds (100 pM, 30 s), and further passivated with casein (Hammarsten Bovine, 5 mg/ml, 10 min) and BSA (50 mg/ml, 10 min). The chamber was finally connected to the microfluidic tubing and thoroughly rinsed with F-buffer. Following buffers were used in the microfluidic studies: F-buffer (5 mM Tris·HCl pH 7.4, 50 mM KCl, 1 mM MgCl2, 0.2 mM EGTA, 0.2 mM ATP, 10 mM DTT, and 1 mM 1,4-Diazabicyclo[2.2.2]octane (DABCO)]) and G-buffer (5 mM Tris·HCl pH 7.4, 0.1 mM CaCl2, 0.2 mM ATP, 1 mM DTT, 0.01% NaN3). Buffers were adapted for experiments with ATP-ATTO by removing ATP, and with gelsolin by adding 0.4 mM CaCl2.

**Image acquisition.** The microfluidic chamber was placed on a Nikon Eclipse Ti inverted microscope, with an oil-immersion 60x objective and optional 1.5x additional magnification. The sample was illuminated with either epifluorescence (X-cite Exacte, Lumen Dynamics), Hilo, or TIRF (ILAS2, Gataca Systems). Images were acquired by an sCMOS Orca-Flash4.0 camera (Hamamatsu). Please note that depending on the passivation quality, fluorophores can bind nonspecifically to the surface and increase the background fluorescence. As a result, the same illumination power and time were not used in all experiments.

**Barbed end polymerization experiments.** Filaments were polymerized from spectrin-actin seeds with about 1 μM G-actin and 1 μM profilin in F-buffer. Rabbit α-skeletal muscle actin was fluorescently labeled with 10–12% Alexa. We employed three different strategies to visualize LmActin:

– Mixed filaments: unlabeled Lm-G-actin was mixed with Rb-G-actin, labeled with 30–40% Alexa, resulting in 10–20% Rb- / 80–90% Lm-F-actin (2.5–5% final labeling fraction). The final fraction of RbActin was estimated based on the fluorescence signal, normalized by that of a 100% RbActin filament with the same labeling fraction.
– ATP-ATTO: 100% LmActin with a fluorescent nucleotide were directly polymerized from spectrin-actin seeds. See the ATP-ATTO labeling section for more details.
– Segmented filaments: Alexa-labeled RbActin and unlabeled LmActin were injected using two distinct microfluidic input channels. Seeds were exposed sequentially to the two polymerizing solutions. LmActin was then detected as segments between two fluorescent RbActin segments.

**ATP-ATTO labeling.** We followed the method by ref. [44] to fluorescently label actin with ATP-ATTO-488 (Jena Bioscience, NU-805–488). To polymerize filaments from spectrin-actin seeds in microfluidics, we mixed 0.8 μM G-actin and 0.8 μM profilin with up to 4 μM ATP-ATTO-488 in F-buffer without ATP. For polymerizing filaments in solution, we proceeded in two steps to let the nucleotide exchange before rapid spontaneous nucleation. We prepared first a solution containing 2 μM actin, 2 μM profilin, and 10 μM ATP-ATTO-488 in G-buffer without ATP, left at room temperature for 30 min for nucleotide exchange. It was then mixed with a solution of an equal volume containing 2 μM G-actin in an F-buffer without ATP and adjusted at 100 mM KCl. The final solution was left at room temperature for over 30 min to complete polymerization.

**Estimation of the polymerization rate.** We used three different methods to measure the polymerization rate depending on the assembly method:

– Fluorescent filaments (Alexa-RbActin, mixed filaments): the polymerization was imaged continuously in TIRFm and the polymerization rate was measured manually as the slope on kymographs (ImageJ).
– Segments: if actin was unlabeled (segmented filaments) or the background fluorescence was too high (ATP-ATTO), the polymerization rate was simply measured as the length of the actin segment divided by the polymerization duration.
– Polymerization/depolymerization with twinfilin: we compared the length of segments assembled for *n* minutes with actin-profilin alone, with or without extra *m* min of actin-profilin and twinfilin. The polymerization rate with twinfilin was calculated as the difference in length, divided by the duration *m* (Fig. 7a).

In all cases, the elongation rate was normalized from μm/s to subunits/s by dividing by the effective monomer length of 2.7 nm. Individual measurements were finally averaged and plotted with the Python package Numpy and Panda.

**Barbed end depolymerization and Pi-release.** The depolymerization rate of RbActin and LmActin was measured by exposing filaments to F-buffer only. To measure the depolymerization rate of ADP-Pi F-actin, F-buffer was modified by replacing KCl with 40 mM $K_2HPO_4$ and 10 mM $KH_2PO_4$. The depolymerization of ADP-LmActin was measured on segments of filaments that have been polymerized for at least 5 min (such that >99% of monomers had released their Pi). Finally, to measure the Pi-release rate, F-actin was polymerization with 0.5–1 μM actin-profilin in a modified F-buffer in which KCl was replaced by 40 mM $K_2HPO_4$ and 10 mM $KH_2PO_4$, yielding filaments in a fully ADP-Pi state. Filaments were then exposed to a standard F-buffer (without the excess of Pi) from $t = 0$. They then started depolymerizing with a rate that depended on the Pi content (faster for ADP-actin than ADP-Pi), while Pi was itself released over time[87]. Movies were analyzed by first making kymographs of the depolymerization (ImageJ) and manually measuring the instantaneous polymerization rate at multiple time points (60 to 80 points per movie). This depolymerization rate vs time v(t) was then fitted on Python (curve_fit function from Scipy package) with the following function[87]:

$$\frac{1}{v(t)} = \frac{1}{v_{ADP}} + e^{-t*k_{off}}\left(\frac{1}{v_{ADP \cdot Pi}} - \frac{1}{v_{ADP}}\right) \qquad (1)$$

where $v_{ADP}$ and $v_{ADP \cdot Pi}$ are the depolymerization rates of purely ADP and ADP·Pi actin filaments, and $k_{off}$ is the Pi-release rate. These three constants are always left as free parameters. Supplementary Fig. 2b, c show the fitted parameter for two typical movies, with LmActin and RbActin, labeled with ATP-ATTO or Alexa568, respectively.

**Pointed end depolymerization rate.** The microfluidic chamber was passivated with 1% biotinylated PLL-PEG or BSA following the previous method. The surface was then exposed to 5 μg/mL neutravidin for 5 min, and 10 nM biotin-gelsolin in F-buffer supplemented with 0.4 mM CaCl2 for ~2 min. F-actin was then injected into the chamber. Gelsolin binds to the side of rabbit and LmActin filaments, severs them, and remains bound to the new barbed end. Filaments were finally exposed to F-buffer to trigger pointed end depolymerization.

**Filament severing rate by LmCofilin.** Actin filaments were exposed from $t = 0$ onwards to a constant concentration of LmCofilin (WT or D4C mutant). For each movie, 20 to 60 filaments were selected unbiasedly on the first frame. For segmented filaments, all unlabeled LmActin segments were then analyzed. For ATP-ATTO or Alexa actin, each filament was divided into segments of fixed length (typically 2 μm for LmActin, 10 μm for RbActin). We then tracked on which frame each segment was either severed or lost (for example because of a severing event on another part of the filament or at the spectrin-actin seed). We then used the Kaplan-Meier estimator to calculate the fraction of severed actin segments (Python packages Numpy and Lifelines). Likewise, we used the "exponential" Greenwood formula to calculate the 95% confidence interval (shown as shaded surfaces)[88]. The fraction curve was fitted (Python function curve_fit, in scipy package) with a single exponential:

$$F(t) = 1 - e^{-k_{sev}*L*t} \qquad (2)$$

where $L$ is the mean segment length and $k_{sev}$ is the global severing rate (unit: /s/μm, Figs. 5b, 6e and Supplementary Fig. 5).

Likewise, the lower and upper values of the 95% confidence interval were fitted to estimate error on the fit (error bars, Figs. 5b, 6e and Supplementary Fig. 5).

**Cryo-EM sample preparation**. For preparing LmActin for plunge-freezing, the ADP-Pi sample was polymerized in PBS supplemented with 0.2 mM ATP, 0.4 mM EGTA, 1 mM MgCl2, and 1 mM DTT for one hour. Actin filaments for freezing with cofilin were polymerized in 10 mM HEPES, 125 mM NaCl, 5 mM KCl, 0.2 mM ATP, 0.4 mM EGTA, 1 mM MgCl$_2$, 1 mM DTT, pH 7.4. The ADP-actin sample was aged 1–2 h in the buffer above before adding cofilin and freezing, and thus we expect that virtually all filaments had reached ADP-status (please note that Pi-release from LmActin filaments is very fast compared to e.g., rabbit muscle actin; see Fig. 2d). All samples contained ~10% remaining of the G-buffer. Three microliters of *Leishmania* actin at 12 µM concentration was applied on a glow discharged Quantifoil 300 mesh copper grid (R1.2/R1.3) and left for 15 s. The grid was plunge-freezed in liquid ethane after blotting for 5 s using a Vitrobot IV operated at a relative humidity of 95%, 6 °C with a blot force of 15. For the actin-cofilin complex, 3 µl of 12 µM *Leishmania* actin was applied on the grid and left to settle for 15 s at room temperature. This grid was then mounted to the Vitrobot and 1 µl of the sample was withdrawn. On top of the remaining 2 µl of *Leishmania* actin, 1 µl of *Leishmania* cofilin at 77 µM concentration was added and plunge-freezed immediately. The grids were stored in liquid nitrogen.

**Cryo-EM data collection**. Data for ADP-Pi actin filament was collected on a Talos Arctica (Thermo Scientific) operated at 200 kV with a C2 aperture of 50 µm and an objective aperture of 100 µm. Movies were recorded at a nominal magnification of 150,000x corresponding to a calibrated pixel size of 0.97 Å yielding a total dose of ~45 e/Å² fractionated into 45 frames. The target defocus for acquisition ranged from −0.8 to −3 µm. The data were collected using the EPU software (Thermo Scientific) with the Falcon 3 detector (Thermo Scientific) in counting mode in two separate sessions yielding 1998 movies in total. Data for ADP-actin filament (with or without cofilin-bound) was collected on a Titan Krios G2 (FEI Thermo Scientific) equipped with a BioQuantum/K3 energy filter (Gatan) operated at 300 kV in zero-loss mode (slit width 20 eV) with a C2 aperture of 50 µm and an objective aperture of 100 µm. Movies were recorded at a nominal magnification of 105,000x corresponding to a calibrated pixel size of 0.86 Å and a total dose of ~55 e/Å² fractionated into 50 frames. The target defocus for acquisition ranged from −0.75 to −2.5 µm. The data were collected using the EPU software (Thermo Scientific) with the detector in counted super-resolution mode using a binning factor of 2. The total amount of movies collected was 8138.

**Image processing**. Cryo-EM data were processed in Relion 3.1 software[89]. Movie frames were first aligned with MotionCor2[90] or Relion's own implementation of motion correction and CTF estimated by CTFFIND 4.1[91]. Filaments were picked manually in Relion for ADP-Pi, ADP-actin, and cofilin-decorated filament reconstructions. For the ADP-Pi sample, manually picked segments were used as templates to automatically pick the final particles from two data sets collected on different days. ADP-actin filament was reconstructed from the same sample as cofilin-decorated actin filaments. This was possible since we observed that only ~5–10% of the images contained filaments that were decorated with cofilin at specific grid locations. This was likely due to the addition of cofilin to actin filaments without mixing just before plunge-freezing. Mixing cofilin homogeneously before plunge-freezing led to completely disintegrated filaments. For helical processing of the filaments, actin filaments were extracted in ~248 Å boxes with a 56-Å inter-box distance, roughly corresponding to two asymmetric units in the filament. To remove bad particles, we first extracted 2x binned segments and carried out 2D classification. Classes resulting in 2D class averages with clear features (bare actin filaments or cofilin-decorated actin filaments) were selected and re-extracted without binning and processed separately. Unbinned segments were classified again in 2D to remove bad particles. For the first 3D refinement, an ab initio model was generated in Relion. Helical parameters were initially searched with a wide search range, and during subsequent refinement steps, narrowed near the converged helical parameters. Several rounds of per-particle CTF refinement and Bayesian polishing steps using a soft-edged solvent mask (z-length 90% of box size) improved the map resolution initially by 0.5–1 Å. One more 3D classification was performed without image alignment to remove poorly aligning particles using a reference model (low-pass filtered to 40 Å) and a soft-edged solvent mask (low-pass filtered to 15 Å, z-length 90% of the box size). Particles that had moved during the processing steps closer to 50 Å of each other were removed. For the remaining particles, polishing and CTF refinement were repeated to produce the particle stack for the final 3D reconstruction. The final map resolution was estimated with a soft cylindrical mask in Relion (z-length of 50%). Sharpening B-factor was automatically estimated by Relion. Local resolution of the cryo-EM reconstructions was estimated with Relion. Finally, we applied helical symmetry on the map in real space using the highest resolution region (z-length of 50%) by using the relion.-helix.toolbox. This map was input to AutoSharpen in Phenix[92] (v. 1.19.2) to produce the primary map for model building.

**Model building**. To build ADP-state *Leishmania* actin filament, the asymmetric unit from chicken muscle actin (PDB ID: 6DJO) was manually docked into the cryo-EM density in Chimera[93] (v. 1.14 build 42094). To transform actin monomer to *L. major* actin sequence, the sequence was automatically assigned in Phenix with the "Map-to-model" function. After mutation to *L. major* sequence, the model was manually

curated and built per amino acid in Coot[94] (v. 0.8.9.1) and refined in Phenix (v. 1.19.2). In all cases, a helical assembly consisting of five actin monomers was built in Chimera. ADP-Pi state was built based on the ADP-state model and flexibly fitted to the cryo-EM map in Namdinator molecular dynamics webserver (https://namdinator.au.dk)[95]. The initial model for LmCofilin was obtained from *Leishmania donovani* cofilin (PDB ID: 2kvk)[96], which was manually mutated to *L. major* cofilin and then docked to the cryo-EM map in Chimera. Then a monomer from LmADP-actin reconstruction was docked to the cryo-EM density and relaxed to LmCofilin-bound conformation in Namdinator. After these steps, the models were manually curated in Coot. Ramachandran and side-chain rotamer outliers were fixed by using the iSolde[97] (v. 1.1.0) package in ChimeraX (v.1.1.1). Several rounds of these actions were performed and refined in Phenix using reference model restrains from the actin monomer (with or without cofilin) lying at the center of the filament. For the ADP-actin model, we also built water molecules where density and water coordination was sensible. The coordinating waters for the central Mg²⁺ ion in the same plane showed clear density, but the fifth water (perpendicular to the plane) did not. We still placed it to the model to complete the coordination and restrained the coordinating waters in the Phenix refinement. The density for the last C-terminal residue F375 was ambiguous in all models. In the ADP-actin model, we placed it to the most likely orientation where we observed some density, in other models we did not have the confidence to build it. D-loop was unstructured in the cofilin-decorated actin model. In the two other models, the density for D-loop residues 46–48 was fragmented but traceable with a lower isosurface threshold.

**Model analysis**. Models were analysed in Pymol (v. 2.3.0) and superimposed in most cases using all main-chain atoms, except for Fig. 5e where the conformational change of cofilin binding is shown. For this comparison, we used main-chain atoms selections for subdomains 3 and 4. The reported RMSD values were calculated using the "Super" function in Pymol (v. 2.3.0).

**Statistics and reproducibility**. In all figures, *n* describes the number of filaments or segments analyzed, whereas *N* describes the number of independent experiments.

**Reporting summary**. Further information on research design is available in the Nature Research Reporting Summary linked to this article.

## Data availability

The cryo-EM reconstructions generated in this study of the *L. major* ADP-actin filament, ADP-Pi-actin filament, and ADP-actin filament–cofilin complex have been deposited in the Electron Microscopy Data Bank (EMDB) under ID accession codes EMD-13864, EMD-13863, and EMD-13865, respectively. The corresponding atomic models generated in this study have been deposited in the Protein Data Bank (PDB) under accession codes 7Q8C, 7Q8B, and 7Q8S, respectively. The coordinates corresponding to the muscle actin filament structures, cofilin-decorated actin filament structure, yeast cofilin, and malaria actin filament structure shown in this article are available from PDB under accession codes 6DJO, 6DJN, 5YU8, 1QPV, and 6TU4, respectively. The sequence data used in this study was obtained from Uniprot and the accession codes are described in the Source Data file. Source data for the biochemical experiments using TIRFM data are provided in Source Data File. Other data are available from the corresponding author upon reasonable request. Source data are provided with this paper.

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

## Acknowledgements

We thank Benita Löflund and Pasi Laurinmäki (University of Helsinki) for technical assistance from the HiLIFE Cryo-EM unit at the University of Helsinki, a member of Instruct-ERIC Centre Finland, FINStruct, Biocenter Finland, and Finnish Cryo-EM bag (CEM00362). We also thank Dustin Morado and Cryo-EM Swedish National Facility funded by the Knut and Alice Wallenberg, Family Erling Persson and Kempe Foundations, SciLifeLab, Stockholm University and Umeå University. Andrea Vizcaino-Castillo is acknowledged for critical reading of the manuscript. The model in Fig. 7c was created with BioRender.com. This work was supported by Jane and Aatos Erkko Foundation (4708679) and the Academy of Finland (grant 302161) to P.L., ERC (grand StG-679116) to A.J., and the ANR (grant RedoxActin) to G.R.-L.

## Author contributions

P.L. provided conceptualization of the project and A.J., J.T.H., P.L., and G.R.-L. supervised the study. T.K. and M.S. conducted the cryo-EM experiments and H.W. performed the biochemical experiments. T.K., K.K., and L.A. cloned, expressed, and purified the proteins. A.J., T.K., P.L., G.R.-L., and H.W. wrote the manuscript with input from all authors.

## Competing interests

The authors declare no competing interests.
