## [Peer Review File · Nature Communications]

REVIEWER COMMENTS

Reviewer #1 (Remarks to the Author):

The manuscript by Kotila et al is an existing piece of work that provides a set of novel data on the actin-driven machinery that is responsible for the Leishmania parasite motility. I enthusiastically support publication of this work in Nature Communications journal once the authors address the following questions/suggestions:

1. Why in Fig 1f polymerization rate at 1 μ M has such a large divergence in comparison with 0.3-0.5 μ M range of LmActin:Profilin?
2. Was there any difference in the rates of elongation in segmented and mixed filaments experiments shown in Fig 1d?
3. What was the rate of elongation and corresponding critical concentration for the pointed ends of the LmActin filaments? What was the reason to omit these experiments if any?
4. In the depolymerization experiments showed in Fig 2 it is unclear what type of mixed filaments were used – were those segmented filaments of mixed filaments?
5. What does the FSC range of 120 in Angstroms mean in Fig S3a? Fourier Shell Correlation has values from 0 to 1 (or 0% to 100%) and reflects the correlation between the FS as a function of spatial frequency.
6. Examples of actual EM images for the ADP-Pi and ADP actins would be appropriate in the Fig. S3.
7. I would strongly encourage the authors to show regions of actual electron density maps and corresponding atomic models not only for the nucleotide binding pocket, but for all the contacts along and between the F-actin's strands, especially at the regions which show deviations from the rabbit muscle actin structure (e.g. D-loop and SD4). This would strengthen the structural conclusions by directly demonstrating the resolution at those crucial regions.
8. F-actin has variable twist but has fixed axial rise. The reported 0.6 Å difference in the axial rise between ADP-Pi and ADP LmActins is quite dramatic. Nevertheless, it is not clear what would be the mechanism by which the dissociation of Pi would result in such an extension of the axial rise. Is it possible that the difference in the scale between the two instruments (Talos vs Krios) introduced the difference between the two actins?
9. I would suggest the authors to move Fig 3a and b to supplementary, while merging Fig 3 c-f and fig 4. It would be easier for the readers if the authors present 5 panels on such a figure showing comparison of major actin interfaces from rabbit skeletal and LmActins.
10. The authors' conclusions regarding the differences between Rb and Lm actins regarding the D-loop interaction with the upper subunit are not well explained. It is not clear why a more prominent hydrophobic interaction of the D-loop with the upper hydrophobic cavity of the SU above it would still result in the weakly polymerizing actin. Maybe it make sense to write a broader summary of the impact of each interface on the stability of the filament (which may be different for each of the 5 regions) to make it more clear for a broader audience? I would encourage authors to explain in more details the mechanism of faster Pi release based on the observed differences in the nucleotide coordinating loops.
11. The twisting of the two domains of actin around the hinge region by cofilin has been reported by the Egelman group in 2011 (Galkin et al., PNAS 2011). This work should be cited along with Tanaka et al. which was published later.
12. It is very intriguing that LmCofilin does not sever mammalian actin and vice versa. I would like to ask the authors to better explain why mouse Cofilin cannot bind LmActin – what part of the LmActin prevents the interaction?

Reviewer #2 (Remarks to the Author):

The authors of the manuscript NCOMMS-21-48144-T report the elucidation of the cryoEM structures of the evolutionary diverged L. major F-actin along with associated proteins. The authors show that in spite of the high degree of variance with mammalian actin, the structures are essentially identical and can co-polymerise. The organisms differ in their actin associated protein that participate in actin turnover and the manuscript details the differences in their turnover rates and use the cryoEM structures to provide a rationale. The manuscript is written in a lucid manner and is accepted for publication after the authors address the following queries:

1. In the microfluidics experiment result shown in Figure 1d, the actin filaments align parallel to the flow direction and no actin filament is visible in any other direction. Did the filaments grow in one direction only ?
2. The three panels of Figure 1d show the same magnification, but the right panel figure shows fewer filaments. The right panel also appears to be more grainy, as compared to the left. Is it just the illumination effect ?
3. The resolution of the structures are reported to be 2.7 – 3.3 Å, how did the authors differentiate the ADP and ADP-Pi states ?
4. The authors use differences in the sequence and fold of D-loop and H-plug for providing the molecular basis for the observed differences. The superposition figures (Figure 3c, 3d) show that the loops adopt a similar conformation, especially in the ADP-Pi state (Fig. 3d). Could the authors quantify the differences, either in the text or in the figures ?

Reviewer #3 (Remarks to the Author):

The manuscript by Kotila et al used TIRFM-based single molecule assays to examine the polymerization/depolymerization kinetics of Leishman and rabbit skeletal actin in the presence and absence of cofactors. The biochemical assays were complimented with CryoEM studies of Leishmania actin to provide a structural insight to their findings. The overall quality of the individual TIRFM and CryoEM experiments are excellent. Being said, the model proposed as to why Leishmania has fewer proteins controlling its dynamics was derived from the TIRFM assays. While that CryoEM images were beautiful, the discussion around these images seems speculative without mutagenesis and reevaluation of the impact of such mutations on actin dynamics in the TIRFM assay, as was done with the D4C cofilin. Regardless, the structural information in the CryoEM images will be valuable to those interested in developing actin inhibitors. The majority of my criticisms are editorial in nature.

- 1) The introduction provided much information about actin that is excellent for a specialist, but not applicable to a broad audience. I would suggest removing information that is not relevant to the study and highlighting the question(s) that you are addressing.
- 2) It is unclear why specific species were selected for comparison in Fig 1b. The rationale should be justified. *Oryctolagus cuniculus* should be added to the table, as this was used in the TIRFM assays.
- 3) The authors did a great job to head off potential criticism of artefacts in their experimental design by looking at the effects of the ATTO-labeling strategy on polymerization kinetics. I appreciate the data showing that ATTO-ATP slowed the rate of polymerization by ~20%, but it is unclear why the same experiment in Fig 1f was not performed with the ATTO-labeled actin. Many of the subsequent assays were performed with ATTO-ATP due to necessity when examining depolymerization.
- 4) In consideration of the model, all of the rates are shown in a single direction. Is this truly the case, or should each of these be shown as equilibria?

Minor comments:

P. 5 Line 12: "We measured a barbed end depolymerization rates of..."

Figure 2d: units are "(/s)". Should be (1/s) or s-1?

Fig 1f: units are "(sub./s)". Others are "(sub/s)"

P. 1 Line 21: "cause 20.000 – 50.000 death". Should (.) = (,)?

Large numbers in methods were written in varying formats: "10 000 – 15 000 psi" and "38 759 x g" vs "59.000 rpm" and "41943.77 daltons"

Figs 5b and 6 e: units are "(/s/um)" Is this correct?

REVIEWER COMMENTS

Reviewer #1 (Remarks to the Author):

The manuscript by Kotila et al is an existing piece of work that provides a set of novel data on the actin-driven machinery that is responsible for the Leishmania parasite motility. I enthusiastically support publication of this work in Nature Communications journal once the authors address the following questions/suggestions:

We thank the reviewer for positive feedback and excellent suggestions for improving the manuscript.

1. Why in Fig 1f polymerization rate at 1 μM has such a large divergence in comparison with 0.3-0.5 μM range of LmActin:Profilin?

This is a good point. As explained in our manuscript, LmActin spontaneously and rapidly nucleates, and this subsequently decreases the concentration of G-actin and the effective actin polymerization rate (comparison is shown in the Supplementary Fig. 1c-d). We used an equimolar concentration of profilin to limit this problem. Yet, we think that 1 μM LmActin:LmProfilin can still have some level of spontaneous nucleation that would account for the measured divergence. This is now clarified in the legend to Fig. 1.

2. Was there any difference in the rates of elongation in segmented and mixed filaments experiments shown in Fig 1d?

To address this point, we performed new experiments to accurately compare the elongation rates of unlabeled LmActin, either pure or when mixed with rabbit Alexa488-actin. We found that a solution of 0.5 μM actin:profilin, with 10 or 20% rabbit actin, polymerizes more slowly than pure solutions of either rabbit actin:profilin or LmActin:LmProfilin. This new result is shown as a new panel (f.) in Supplementary Fig. 1, and is discussed in the 'Results' on page 4 lines 23-29. The slower polymerization of mixed actin is consistent with the faster barbed end depolymerization of mixed filaments (Supplementary Fig. S2a) and the idea that LmActin subunits and rabbit actin subunits display nonoptimal interactions with each other.

3. What was the rate of elongation and corresponding critical concentration for the pointed ends of the LmActin filaments? What was the reason to omit these experiments if any?

We agree with the reviewer that measuring the pointed end elongation rate and critical concentration of LmActin would be very interesting information, and we considered executing such experiments during the study. Unfortunately, such information is very difficult to obtain with our current experimental setup. Given the slow polymerization rate of filament pointed ends, we would have to use very high concentrations of G-actin ($>10 \mu\text{M}$) to observe filament pointed end elongation in our setup. Furthermore, these experiments would have to be conducted in the absence of profilin, because profilin prevents pointed end elongation. As discussed above, at such concentrations LmActin would spontaneously nucleate extremely rapidly and thus deplete the G-actin stock, and hence prevent reliable analysis of the pointed end elongation rate. This is now discussed on page 4. Nevertheless, we speculate that the critical concentration of LmActin pointed ends is probably very high due to the observed ~ 20 -fold faster off-rate of Lm-ADP-actin monomers compared to rabbit muscle actin.

We would also like to point out that, in the experiments for measuring the pointed end depolymerization rate (Fig. 2e-f), filaments were not polymerized inside the microfluidic chamber. Instead, they were pre-polymerized in solution and then injected into the microfluidic chamber, where they were captured by surface-anchored gelsolin.

4. In the depolymerization experiments showed in Fig 2 it is unclear what type of mixed filaments were used – were those segmented filaments of mixed filaments?

The actin filaments shown in kymographs in Fig. 2a for both species are labelled with ATP-ATTO-488. The kymographs shown in Fig. 2e for RbActin filaments were detected with Alexa488-labelled RbActin (10-12% fraction), whereas the LmActin filaments were labelled with ATP-ATTO-488. We have now annotated the kymographs accordingly. The depolymerization data for Leishmania actin shown in panels Fig. 2b-d and Fig. 2f do not contain data obtained with 'mixed filaments', because the presence of rabbit actin affects the filament depolymerization rates. These data are separately shown in Supplementary Fig. S2. In these experiments shown in Supplementary Fig. S2, the "mixed filaments" were homogeneous filaments, polymerized from a single solution containing a mix of unlabeled LmActin and 9-20% Alexa-labeled rabbit actin. This is now clarified in the figure legend and 'Methods'. Please note that the data for barbed end depolymerization of segmented filaments are also shown in Supplementary Fig S2a.

5. What does the FSC range of 120 in Angstroms mean in Fig S3a? Fourier Shell Correlation has values from 0 to 1 (or 0% to 100%) and reflects the correlation between the FS as a function of spatial frequency.

We apologize for the confusing representation in this figure. The number '120Å' indicates the z-length of the mask that was applied to calculate the final reported resolution. For clarity, we have removed this number from the figure. We have described in the 'Methods' (page 24, line 2) how the final resolution of reconstructions were determined.

6. Examples of actual EM images for the ADP-Pi and ADP actins would be appropriate in the Fig. S3.

We have added examples these EM images to Supplementary Fig. S3a.

7. I would strongly encourage the authors to show regions of actual electron density maps and corresponding atomic models not only for the nucleotide binding pocket, but for all the contacts along and between the F-actin's strands, especially at the regions which show deviations from the rabbit muscle actin structure (e.g. D-loop and SD4). This would strength the structural conclusions by directly demonstrating the resolution at those crucial regions.

We have followed the reviewer's suggestion and included a new supplementary figure (now Fig. S4), which shows the cryo-EM densities of the amino acid residues in the contacting regions of the actin subunits within the filament. As shown by the figure, majority of the side chains can be reliably built into the density. We would also like to emphasize that we have not made any strong conclusions from regions with poor or nonexistent cryoEM density.

8. F-actin has variable twist but has fixed axial rise. The reported 0.6 Å difference in the axial rise between ADP-Pi and ADP LmActins is quite dramatic. Nevertheless, it is not clear what would be the mechanism by which the dissociation of Pi would result in such an extension of the axial rise. Is it possible that the difference in the scale between the two instruments (Talos vs Krios) introduced the difference between the two actins?

We agree with the reviewer that the difference in axial rise between ADP-Pi and ADP LmActins could be due to the two different microscopy setups. Thus, we do not want to make any strong claims of this difference. We have accordingly modified this section in the 'Results' on page 6.

9. I would suggest the authors to move Fig 3a and b to supplementary, while merging Fig 3 c-f and fig 4. It would be easier for the readers if the authors present 5 panels on such a figure showing comparison of major actin interfaces from rabbit skeletal and LmActins.

We have considered this suggestion and agree that the figures should be easy to interpret also by non-specialist readers. However, we would like to argue that these panels (Fig. 3a and 3b) represent very central results of the manuscript. In our opinion, it is important to show the cryo-EM reconstructions in the main figure, because they represent cryoEM structures that were determined in this study, and which form the framework for subsequent structural analysis. This is especially important for non-specialist readers so that the structural models shown afterwards will not be mistakenly thought to present computational models or other theoretical models of the actin monomers/filaments from Leishmania species.

We also considered combining Figs. 3c-f and Fig. 4 into a single figure, but this would be quite challenging due to the size restrictions of the figures. We hope the reviewer will understand our views on this. However, if this is considered absolutely necessary, we can of course make these changes to Figs. 3 and 4.

10. The authors' conclusions regarding the differences between Rb and Lm actins regarding the D-loop interaction with the upper subunit are not well explained. It is not clear why a more prominent hydrophobic interaction of the D-loop with the upper hydrophobic cavity of the SU above it would still result in the weakly polymerizing actin. Maybe it make sense to write a broader summary of the impact of each interface on the stability of the filament (which may be different for each of the 5 regions) to make it more clear for a broader audience? I would encourage authors to explain in more details the mechanism of faster Pi release based on the observed differences in the nucleotide coordinating loops.

We thank the reviewer for this suggestion, and have now edited the manuscript text (pages 6-7) to more thoroughly summarize the possible roles of each interface (including the D-loop interaction with the upper subunit) on the stability of LmActin filaments, as well as modified Fig. 4 for better clarity.

We would also like to point out that the role of D-loop is open to different interpretations due to several reasons. For example, a recent MD simulation study (Zsolnay et al. 2020, PNAS) and earlier crosslinking study (Scoville et al 2006, Biochemistry) demonstrated that vertebrate actin D-loop can also make interactions with H-plug of the lateral subunit at the pointed end. Furthermore, the conformational landscape of D-loop seems to play an important role in the filament stability (Matsuzaki et al 2020 Biomolecules; Chu et al. 2005 PNAS; Durer et al 2012 Biophys. J). Finally, based on our structure the D-loop of LmActin appears display more prominent hydrophobic interactions with upper subunit, and these could possibly also compensate for the weaker lateral subunit interactions of LmActin. We have now included these speculations to the 'Results' on page 7, and in 'Discussion' on page 12.

We have also added more discussion about the possible mechanism of faster phosphate release of LmActin on page 8. Please note that the mechanism of phosphate release in actin filaments is still incompletely understood. We did not observe alterations in the sequence of the proposed 'backdoor' pathway for phosphate release between Leishmania and vertebrate actin. Thus, most likely, the dynamics of the nucleotide binding region, and the interactions between the individual actin subunits in the filament, affect the accessibility of the phosphate to the 'backdoor' pathway.

11. The twisting of the two domains of actin around the hinge region by cofilin has been reported by the Egelman group in 2011 (Galkin et al., PNAS 2011). This work should be cited along with Tanaka et al. which was published later.

We apologize for not citing this important work in the first version of our manuscript. We have now cited the article accordingly.

12. It is very intriguing that LmCofilin does not server mammalian actin and vice versa. I would like to ask the authors to better explain why mouse Cofilin cannot bind LmActin – what part of the LmActin prevents the interaction?

*This is an excellent question! As shown in the Supplementary fig. 5, the cofilin-binding interfaces of LmActin and vertebrate actin are well conserved. However, the D-loop sequence makes an exception in this case. A likely option is that that mouse cofilin N-terminus is not suitable for interactions with the D-loop of LmActin. As shown by various studies, the N-terminus of cofilin is critical for actin filament binding, and harbors a phosphorylation site that controls the actin filament binding. Another option is that mouse cofilin would require the extended C-terminus (as in LmCofilin) for interactions with LmActin to introduce conformational change actin and achieve high affinity binding. This is suggested by a study of *C. elegans* cofilin (Ono et al. 2001 JBC). These two possible explanations are now discussed in the manuscript in the 'Results' and 'Discussion' section, on page 10 and 13, respectively.*

Reviewer #2 (Remarks to the Author)

The authors of the manuscript NCOMMS-21-48144-T report the elucidation of the cryoEM structures of the evolutionary diverged L. major F-actin along with associated proteins. The authors show that in spite of the high degree of variance with mammalian actin, the structures are essentially identical and can co-polymerize. The organisms differ in their actin associated protein that participate in actin turnover and the manuscript details the differences in their turnover rates and use the cryoEM structures to provide a rationale. The manuscript is written in a lucid manner and is accepted for publication after the authors address the following queries:

We thank the reviewer for the positive feedback on our manuscript.

1. In the microfluidics experiment result shown in Figure 1d, the actin filaments align parallel to the flow direction and no actin filament is visible in any other direction. Did the filaments grow in one direction only

In our microfluidic experiments, filaments are polymerized from surface-anchored spectrin-actin seeds. They do not interact with the surface otherwise and, thus align with the flow direction, as sketched in Fig 1c. We modified the 'Results' section (page 4) to clarify this point. The experimental method is thoroughly described in our review paper [Wioland J. Muscle Res. Cell Motil. 2020].

2. The three panels of Figure 1d show the same magnification, but the right panel figure shows fewer filaments. The right panel also appears to be more grainy, as compared to the left. Is it just the illumination effect ?

The number of filaments per field of view directly depends on the initial spectrin-actin seed density. One limitation of our setup is that it is challenging to precisely control the seed density. Even when using the same seed concentration and incubation duration (see Method section "Microfluidic experiments"), we regularly get surfaces with twice as much or fewer seeds (the final density likely depends on the glass surface cleaning and passivation, which are often difficult to control in such experiments).

The quality of the images depends on several factors. First, filaments in Fig 1c-d have different labeling fractions (e.g 10% in the labeled part of segmented filaments, versus 3 to 5 % in mixed filaments). Then, depending on the passivation quality, fluorophores can bind nonspecifically to the surface and increase the background fluorescence. This is particularly true with ATP-ATTO-488. As a result, we cannot use the same illumination power and time in all experiments. Finally, we adjusted the contrast on Fig 1d such that all filaments are clearly visible, though some images appear more grainy. This is now explained in 'Methods' on page 19.

3. The resolution of the structures are reported to be 2.7 – 3.3 Å, how did the authors differentiate the ADP and ADP-Pi states ?

ADP-Pi-actin sample was prepared in phosphate-buffered saline, and the cryoEM reconstruction from this condition showed an extra density in the phosphate-binding site. On the other hand, the ADP-actin sample was aged >1 hour in regular F-buffer before freezing, and thus we expect that virtually all filaments had reached ADP-status (please note that Pi-release from LmActin filaments is very fast compared to e.g. rabbit muscle actin; see Fig. 2d). Accordingly, the reconstructions did not show similar extra density in the phosphate-binding site as with ADP-Pi-actin sample. The cryoEM sample preparation is now more thoroughly described in the 'Methods' page 22.

4. The authors use differences in the sequence and fold of D-loop and H-plug for providing the molecular basis for the observed differences. The superposition figures (Figure 3c, 3d) show that the loops adopt a similar conformation, especially in the ADP-Pi state (Fig. 3d). Could the authors quantify the differences, either in the text or in the figures ?

We appreciate the reviewer's suggestion to quantify differences between these regions. We have reported an overall RMSD difference between the vertebrate and rabbit actin structures (p. 6 lines 17 and 20). However, quantifying a meaningful number for the D-loop and H-plug is challenging especially due to different overall resolutions of the reconstructions from Leishmania and muscle actin (2.7 Å vs. 3.6 Å). Because D-loop is flexible (which is also evident from weaker electron density at certain regions of the D-loop in our and the other structures) providing RMSD differences between different D-loops is not particularly informative. This question would be better addressed by MD simulations, which are beyond the scope of this study. Thus, we have modified the 'Results' section (page 7) accordingly, not to emphasize the position of the D-loop as a major determinant in the differences in filament stability, but rather its primary sequence.

Reviewer #3 (Remarks to the Author):

The manuscript by Kotila et al used TIRFM-based single molecule assays to examine the polymerization/depolymerization kinetics of Leishman and rabbit skeletal actin in the presence and absence of cofactors. The biochemical assays were complimented with CryoEM studies of Leishmania actin to provide a structural insight to their findings. The overall quality of the individual TIRFM and CryoEM experiments are excellent. Being said, the model proposed as to why Leishmania has fewer proteins controlling its dynamics was derived from the TIRFM assays. While that CryoEM images were beautiful, the discussion around these images seems speculative without mutagenesis and reevaluation of the impact of such mutations on actin dynamics in the TIRFM assay, as was done with the D4C cofilin. Regardless, the structural information in the CryoEM images will be valuable to those interested in developing actin inhibitors. The majority of my criticisms are editorial in nature.

We thank the reviewer for positive feedback and valuable suggestions to improve the manuscript. We also agree with the reviewer that to fully understand the role of each structural element in the dynamics of LmActin filament, one needs to combine extensive mutagenesis, biochemistry, and MD simulation approaches in the future. Systematic mutagenesis experiments will, however, be quite challenging in the case of LmActin. First, unlike in the case of cofilin (where the D4C mutation compromised the severing activity), one would need to design 'gain-of-function' mutations that increase the stability of LmActin filaments. This would require simultaneous introduction of mutations to two different actin interfaces (which interact with each other within actin filament). Second, the introduction of vertebrate-specific actin segments to (two interfaces of) LmActin may result in unexpected effects e.g. in the local folding, nucleotide binding and hydrolysis, or overall dynamics of actin, making the interpretation of results challenging. We have accordingly edited the 'Results' to describe the structural differences between Leishmania and vertebrate actins in more detail.

We also point out that the first flagellated parasite actin structures (in ADP-, ADP-Pi, and cofilin form), together with detailed biochemical analysis of their dynamics presented here, lay a foundation for future

systematic mutagenesis and MD simulation studies (see 'Discussion' on page 15).

1) The introduction provided much information about actin that is excellent for a specialist, but not applicable to a broad audience. I would suggest removing information that is not relevant to the study and highlighting the question(s) that you are addressing.

We have now done our best to edit the 'Introduction' to make it better accessible for a wider audience.

2) It is unclear why specific species were selected for comparison in Fig 1b. The rationale should be justified. *Oryctolagus cuniculus* should be added to the table, as this was used in the TIRFM assays.

We selected typical model organisms that have been utilized in biochemical studies of actin and are highly divergent in evolution. This is now explained on page 3, and in the legend to Fig. 1. We have followed reviewer's suggestion and now included also the sequence identity of rabbit actin to the table.

3) The authors did a great job to head off potential criticism of artefacts in their experimental design by looking at the effects of the ATTO-labeling strategy on polymerization kinetics. I appreciate the data showing that ATTO-ATP slowed the rate of polymerization by ~20%, but it is unclear why the same experiment in Fig 1f was not performed with the ATTO-labeled actin. Many of the subsequent assays were performed with ATTO-ATP due to necessity when examining depolymerization.

For the initial submission, we preferred to use unlabeled actin whenever possible to avoid artefacts caused by the fluorophores. Following the reviewer's suggestion, we have now performed new experiments to also measure the polymerization rate of ATP-ATTO labeled LmActin at different concentrations. The new experiments confirm that the addition of ATP-ATTO slightly reduces the filament barbed end polymerization rate. Fig. 1f and Supplementary Fig. S1e have been accordingly modified to include the new data.

4) In consideration of the model, all of the rates are shown in a single direction. Is this truly the case, or should each of these be shown as equilibria?

As pointed out by the reviewer, the final model presents majority of the reactions only in one direction, while reverse reactions of course also occur. We made this simplification for three reasons. First, our manuscript focuses on the mechanisms of actin filament disassembly, and therefore these are also highlighted in the model figure. Secondly, for technical reasons it is not feasible to determine some of the reverse rates (e.g. due to rapid spontaneous nucleation of LmActin the rates of actin monomer assembly to the filament pointed ends cannot be reliably determined, and due to very rapid severing of LmActin filaments by cofilin the rates cofilin:actin association and dissociation at filament ends cannot be measured). Finally, we expect that most of the reverse reactions are either very slow (e.g. cofilin unbinding from the sides of filaments, and actin:cofilin binding to the pointed end), or non-existent (e.g. in cells, polymerization at the filament pointed end is inhibited by profilin). This is now clarified in the legend of Fig. 7. We have now also changed the title of Fig. 7 to 'Principles of rapid actin filament disassembly in Leishmania parasites' (previously 'Principles of rapid actin dynamics in Leishmania parasites').

Minor comments:

P. 5 Line 12: "We measured a barbed end depolymerization rates of..."

Thank you for pointing this out. This is now corrected.

Figure 2d: units are "(/s)". Should be (1/s) or s⁻¹?

We have fixed this to 1/s.

Fig 1f: units are "(sub./s)". Others are "(sub/s)"

We have unified these in all figures to sub/s.

P. 1 Line 21: "cause 20.000 – 50.000 death". Should (.) = (,)?

We have deleted this sentence from the edited 'Introduction'.

Large numbers in methods were written in varying formats: "10 000 – 15 000 psi" and "38 759 x g" vs "59.000 rpm" and "41943.77 daltons"

We have corrected these to follow the same format.

Figs 5b and 6 e: units are "(/s/um)" Is this correct?

Yes, this is the number of severing events observed, per second and per μm of actin filament. This is now better explained in the figure legends.

REVIEWERS' COMMENTS

Reviewer #1 (Remarks to the Author):

The authors addressed all my concerns and suggestions. Therefore, I'm happy to recommend the manuscript for acceptance.